# ChainAgile: A framework for the improvement of Scrum Agile distributed software development based on blockchain

**Junaid Nasir Qureshi, Muhammad Shoaib Farooq**[ID]*

Department of Computer Science, School of System and Technology, University of Management and Technology, Lahore, Punjab, Pakistan

* shoaib.farooq@umt.edu.pk

**Data Availability Statement:** The data supporting the findings of this study are openly available in the GitHub repository at the following link: https://github.com/JunaidNasirQureshi/ChainAgile_Blockchain. The minimal dataset containing

## Abstract

Software Development based on Scrum Agile in a distributed development environment plays a pivotal role in the contemporary software industry by facilitating software development across geographic boundaries. However, in the past different frameworks utilized to address the challenges like communication and collaboration in scrum agile distributed software development (SADSD) were notably inadequate in transparency, security, traceability, geographically dispersed location work agreements, geographically dispersed teamwork effectiveness, and trust. These deficiencies frequently resulted in delays in software development and deployment, customer dissatisfaction, canceled agreements, project failures, and disputes over payments between customers and development teams. To address these challenges of SADSD, this paper proposes a new framework called ChainAgile, which leverages blockchain technology. ChainAgile employs a private Ethereum blockchain to facilitate the execution of smart contracts. These smart contracts cover a range of functions, including acceptance testing, secure payments, requirement verification, task prioritization, sprint backlog, user story design and development and payments with the automated distribution of payments via digital wallets to development teams. Moreover, in the ChainAgile framework, smart contracts also play a pivotal role in automatically imposing penalties on customers for making late payments or for no payments and penalties on developers for completing the tasks that exceed their deadlines. Furthermore, ChainAgile effectively addresses the scalability limitations intrinsic in blockchain technology by incorporating the Interplanetary File System (IPFS) is used for storage solutions as an off-chain mechanism. The experimental results conclusively show that this innovative approach substantially improves transparency, traceability, coordination, communication, security, and trust for both customers and developers engaged in scrum agile distributed software development (SADSD).

## Introduction

In recent decades, the adoption of scrum agile distributed software development (SADSD) has seen a significant rise, driven by the increasing demand for global software development [1].

information on latency and size for 700 blocks, as measured during the tests, is available in the Supporting information file 'Dataset.csv'.

**Funding:** The author(s) received no specific funding for this work.;

**Competing interests:** The authors have declared that no competing interests exist.

Distributed Software Development [2] involves collaborative software development efforts from different geographical locations. Agile methodology prioritizes incremental and iterative development, encouraging self-organizing and cross-functional teams to collaborate on refining requirements and software solutions [3]. The advantages of Scrum Agile development include continuous testing for improvements, adaptive planning, quick responses to customer changes, and early delivery of prioritized tasks [4]. The Scrum Agile development process typically encompasses a six-phase lifecycle, as depicted in Fig 1. SADSD essentially combines Scrum Agile software development with Distributed Software Development [5]. SADSD relies on geographically dispersed Agile teams, offering notable benefits like enhanced quality, reduced development timelines, and cost-efficiency [2]. Scrum Agile Distributed Software Development (SADSD) presents significant hurdles for development teams, with a primary focus on achieving effective communication and collaboration [6] in geographically dispersed environments. The geographical distribution challenges primarily encompass concerns related to traceability, communication, security, transparency, remote location teamwork agreements, geographically dispersed teamwork efficiency, and trust [7]. Some recent research has indicated that current communication tools can alleviate some of these challenges [8], but there are still existing deficiencies in areas such as transparency, trust, security, and traceability. These problems can result in project delays, project cancellations, teamwork agreements, product backlog, remote teamwork effectiveness, customer dissatisfaction, and disputes over payments between customers and developers.

In a decentralized distributed environment, both customers and developers encounter substantial obstacles that demand attention to attain project goals. Customers situated in various regions express concerns about the software development process and their satisfaction with the product. Meanwhile, developers may face trust-related issues concerning payments, expenses related to project cancellations, teamwork agreement cancellation, and continuous coordination with the development team. The integration of blockchain technology can serve as a pivotal solution to mitigate these challenges by encouraging trust among distributed customers and developers geographically dispersed across the globe. To tackle the scalability challenge in ChainAgile, an extensive volume of records related to developers, clients, and customers is stored using IPFS. This strategy of data load on the blockchain, results in enhanced transaction performance [9]. However, it's essential to note that all data traversing the six layers of ChainAgile is preserved within the blockchain to monitor the progress of

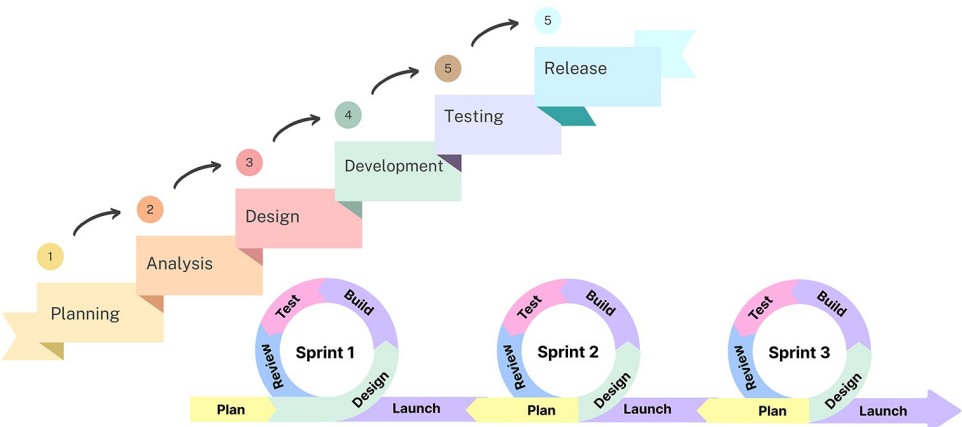

**Fig 1. SDLC scrum agile software development life cycle.**

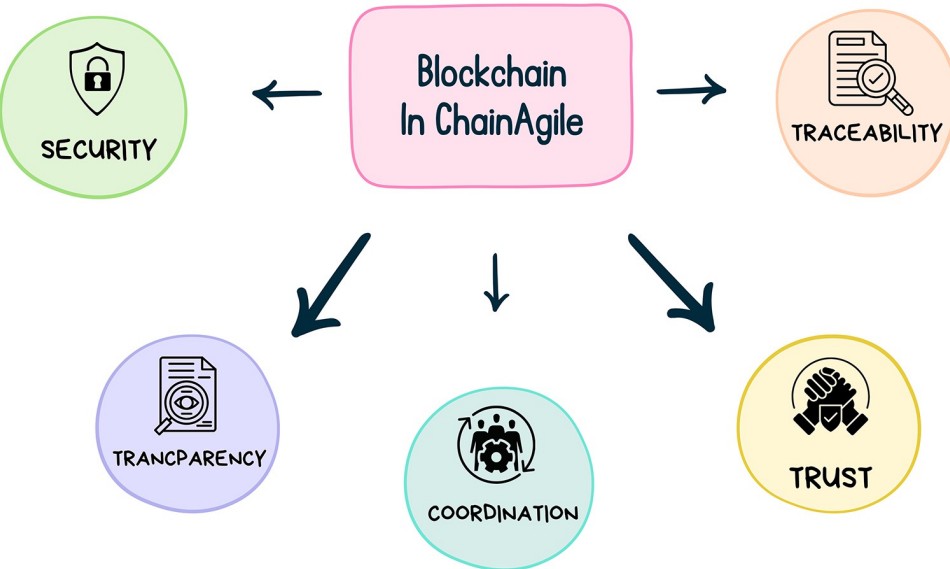

**Fig 2. Blockchain features in ChainAgile.**

(SADSD) agile teams. The principal objective of this study is to leverage blockchain technology to confront the major challenges associated with geographical dispersion in SADSD.

The ChainAgile framework employs the deployment and execution of smart contracts as a strategic approach to address challenges related to communication, trust, security, traceability, transparency, and coordination. Fig 2 illustrates the blockchain features in ChainAgile. In the proposed framework of scrum agile distributed software development (SADSD) the integration of Blockchain provides security, traceability, coordination, trust, and transparency to both client customer and developer teams in geographically dispersed software development environments.

## Motivation

As the decentralized distributed global software development environment, both customers and developers encounter substantial obstacles that demand attention to attain project goals. Customers situated in various regions express concerns about the software development process and their satisfaction with the product. Meanwhile, developers may face trust-related issues concerning payments, expenses related to project cancellations, teamwork agreement cancellation, and continuous coordination with the development team. Therefore, the integration of blockchain technology can serve as a pivotal solution to mitigate these challenges by encouraging trust among distributed customers and developers geographically dispersed across the globe. The advantage of integrating blockchain technology into decentralized systems stems from its potential to revolutionize traditional paradigms. Blockchain's decentralized nature eliminates the need for central authorities, fostering trust and transparency. This technology serves as a catalyst for innovation by providing a secure and tamper-proof ledger. In decentralized systems, blockchain motivates adoption by offering a reliable mechanism for transparent and immutable record-keeping. It addresses trust issues, reduces reliance on intermediaries, and empowers participants with a shared and verifiable source of truth. This motivates and extends across various aspects of distributed software development where the advantages of decentralization and enhanced security are crucial.

## Contribution

The primary contributions of this study include: Proposed ChainAgile Framework for Scrum Agile Distributed Software Development (SADSD), which incorporates a private Ethereum blockchain to enhance security. The framework is further aided by an intuitive DApp interface, promoting efficient communication and collaboration among its users. To mitigate scalability challenges within the blockchain infrastructure the combination of Interplanetary File System (IPFS) provides a secondary storage solution to effectively manage JSON objects. Empowering ChainAgile users to initiate Ether cryptocurrency transactions within digital wallets for payments, with the flexibility to convert these into different fiat currencies through a currency conversion service. The proposed ChainAgile Framework for Scrum Agile Distributed Software Development (SADSD) provides the solution by integrating smart contracts to manage teamwork agreements, product backlog, task prioritization, remote teamwork effectiveness to overcome trust-related issues, project delays, project cancellations, customer dissatisfaction.

- ChainAgile framework provides teamwork agreements, product backlog, task prioritization, remote teamwork effectiveness, and enforcing penalties for users who do not promptly adhere to predefined terms and conditions of software development by integrating smart contracts.

- The deployment and execution of smart contracts within ChainAgile serve to enhance various aspects, including streamlining acceptance testing, ensuring secure payment processing, validating developer payment requirements, and automating the distribution of payments among development teams.

- Evaluating ChainAgile's performance by analyzing the expansion of the blockchain's size with the inclusion of new block transactions and measuring the latency when retrieving the longest chain.

- Complete storage of records across all six layers within the blockchain, ensuring effective tracking of the progress of distributed teams and project development milestones.

- ChainAgile framework integrates blockchain with scrum agile distributed software development (SADSD).

This research introduces a novel approach by integrating blockchain technology into a user-friendly framework to address transparency, security, traceability, and trust issues between customers and developers in SADSD. Previous studies have not proposed such a framework. Smart contracts play a pivotal role in the execution of various tasks, such as acceptance testing, secure payment processing, payment requirement validation, teamwork agreements, product backlog, task prioritization, remote teamwork effectiveness, and the automated distribution of payments. The transactions can easily be converted to any fiat currency with the use of cryptocurrency ether via a currency conversion service. Furthermore, smart contracts are designed to impose penalties in cases where predefined terms and conditions are not met. To tackle scalability concerns, the integration of IPFS is implemented, and JSON objects are utilized as part of the system.

## Blockchain role and its importance

Employing blockchain in a decentralized distributed environment is manifold. Blockchain serves as a foundational technology that enhances the efficiency and reliability of decentralized distributed networks. Its key benefits include heightened security, verification, transparent,

ensuring data integrity, tamper-resistant record-keeping, and the elimination of single points of failure. Blockchain plays a pivotal role in securing and validating digital transactions by utilizing a decentralized and distributed ledger. It ensures transparency, as each participant in the network has access to the entire chain of transactions. The blockchain's importance lies in its ability to establish trust without the need for intermediaries. The benefits of decentralization often include increased resilience, improved security, reduced dependency on a single point of failure, enhanced transparency, and greater autonomy for participants. Smart contracts, automated and self-executing, streamline processes, reducing the reliance on intermediaries and improving efficiency. In the proposed model of SADSD the integration of blockchain encompasses the security, reliability, transparency, efficiency, and streamlined automation overcoming the challenges of coordination and communication in a distributed environment. Furthermore, the geographical distribution challenges primarily encompass concerns related to traceability, communication, security, transparency, remote location teamwork agreements, geographically dispersed teamwork efficiency, and trust-related issues.

## Blockchain integration benefits

Blockchain serves as a foundational technology that enhances the efficiency and reliability of decentralized distributed networks. In ChainAgile framework for scrum agile distributed software development provides key benefits like security, verification, transparency, ensuring data integrity, tamper-resistant record-keeping, automatic payment mechanism, dealing with penalties, and the elimination of single points of failure.

The paper's structure is organized as follows: The related work section discusses relevant prior research, the Materials and Methods section outlines the foundational concepts utilized in the framework introduces the ChainAgile framework, and elucidates its functionality. Discussion Section which expresses the whole article formation. Results Section shows the experiment results, and performance evaluation, and explores the findings. Finally, the conclusion section concludes and outlines potential opportunities for future research.

## Related work

Different frameworks and designed models have been proposed previously to tackle the scrum agile distributed software development (SADSD) challenges for customers, clients, and development teams. Although numerous frameworks have effectively tried communication, collaboration, and coordination challenges, they have not yet furnished a comprehensive solution encompassing trust, traceability, transparency, remote location teamwork agreements, geographically dispersed teamwork efficiency, and security through the integration of blockchain technology.

## Blockchain-based

Despite numerous studies attempting to address challenges in SADSD, only a limited few have explored the potential of blockchain technology in establishing traceability, communication, security, transparency, remote location teamwork agreements, geographically dispersed teamwork efficiency, and trust. For example,

[10] introduced a blockchain-based framework specifically for requirements traceability in the software development lifecycle. This innovative framework eradicates the necessity for tracing requirements, this gives a collaborative environment that enhances the accuracy, and reliability of requirements traceability. However, this framework primarily focuses on the requirements engineering process and may not be ideally suited for the broader scope of

SADSD. Limitations: This only works requirement specification other related issues to distributed scrum agile software development persist.

[11] This study provides a blockchain-based framework and tries to overcome distributed agile development challenges. It did not focus on scrum-based distributed challenges specifically prioritization tasks, product backlog, sprint backlog, and retrospective-review-based task prioritization. This handles and manages the limited transactions. Our innovative proposed framework provides the solution for traceability, communication, security, transparency, remote location teamwork agreements, geographically dispersed teamwork efficiency, and trust, and also the large-scale comprehensive solution to handle the prioritization tasks, product backlog, sprint backlog, and retrospective-review-based task prioritization for scrum agile. Limitations: issues related to distributed scrum agile software development persist because of these prioritization tasks, product backlog, and sprint backlog, these are not addressed.

[12] suggested the use of smart contracts on the Ethereum blockchain to streamline acceptance testing and payment processes within Scrum or Lean-Kanban methodologies. While this approach effectively addresses transparency issues within Lean-Kanban or Scrum, its applicability to distributed environments has some limitations. Furthermore, it does not provide comprehensive solutions for security and traceability issues through blockchain technology in SADSD. Limitations: The model works in a distributed environment but still fails to address the issues related to distributed scrum agile software development.

[13] In this article, a solution is given for smart villages based on the integration of distributed fog computing with IoT-based devices to provide security and privacy. This is an intrusion detection system based on distributed fog computing. This provides security and privacy. The study is not based on distributed software development having Blockchain as an integrated technology and the problems related to software development persist. Specifically, the communication and coordination of offshore development Limitations: The study provides comprehensive security and privacy in a distributed network environment but has limitations related to distributed scrum agile software development network processes to provide specifically the task assignment, its deployment, payments and prioritization tasks, product backlog, and sprint backlog.

[14] This article provides a security network integrated with blockchain technology for smart consumer applications. It provides authentication and security over the network due to the integration of blockchain-based proof of authority. This also provides a deep learning-based intrusion detection system. So this provides a blockchain-based secure network for IoT. The chainagile is a distributed blockchain-based framework that overcomes distributed software development problems by integrating scrum agile with blockchain in a distributed environment. Limitations: The study provides comprehensive security and privacy in a distributed network environment for smart consumer applications and is limited to the distributed scrum agile software development processes.

[15] In this article, a secure data-sharing framework is proposed for Softwarized Unmanned Aerial Vehicles. Blockchain and deep learning-based techniques are used to control network data sharing. The issues of security and vulnerability on the network are controlled due to the use of blockchain to provide proof of authority. ChainAgile provides security, communication, coordination, traceability, transparency, and trust-related issues of distributed software development in Scrum Agile. ChainAgile integrated with blockchain overcomes the distributed software development environment challenges and issues. Limitations: The study provides comprehensive security and privacy also protection for data sharing but is limited in distributed scrum agile software development for its customer and developer communication and coordination specifically the development processes of scrum agile.

[15] In this article digital twin-enabled network is proposed based on a framework integrated with blockchain and deep learning techniques to provide security and privacy. It incorporates IPFS off-chain storage system to store encrypted industrial Internet of things transactions. In comparison the chainagile uses blockchain to provide and overcome the distributed development environment trust-related issues and IPFS off-chain storage is used to store the transactions to provide traceability and transparency. It also integrates a distributed scrum agile development environment with blockchain techniques to provide communication and coordination-related issues. Limitations: The study provides comprehensive security and privacy also IPFS off-chain storage for data sharing but is limited in distributed scrum agile development environment of customer and developer communication and coordination specifically the development processes of scrum agile.

The Blockchain and Deep Learning-based Digital Twin Empowered Industrial IoT Network, Blockchain and Deep Learning-based Framework for Softwarized UAVs, Blockchain-based Authentication for Consumer IoT Applications, and Fog Intelligence for Secure Smart Villages are all sustainable distributed networks. ChainAgile is also a sustainable framework solution in distributed decentralized scrum agile software development with the integration of blockchain in the network using smart contracts to provide immutability, sustainability, transparency, security, trackable, provides secure coordination and communication of a distributed environment.

## Distributed decentralized communication and coordination

[5] Provides an analysis of the communication and performance demands in geographically distributed software development, highlighting how agile methodologies enhance communication through quantitative analysis. Despite these improvements, challenges related to transparency, scalability, and security persist. Limitations: The challenges in the distributed scrum agile development environment of customers and developers related to transparency, scalability, and security persist.

[16] conducted research focused on the communication hurdles encountered in distributed software development. Their investigation explored the potential of collaboration and communication in social media. These findings indicated that the central challenges in SADSD are primarily related to coordination, efficient communication, trust-related issues, cultural differences, and time zone variations. The use of social media helps to address team communication and team coordination challenges, but this does not provide comprehensive solutions to major concerns such as traceability, security, transparency, teamwork agreements, product backlog, task prioritization remote teamwork effectiveness, and trust-related issues. Limitations: The challenges in the distributed scrum agile development environment of customers and developers related to traceability, security, transparency, teamwork agreements, product backlog, and task prioritization persist.

[2] Task allocation framework is described, involving the identification of key factors and dependencies that significantly affect the allocation of tasks to team members based on their compatibility with task requirements. However, this framework doesn't fully address issues of in-person communication, transparency, and scalability, and lacks integration with blockchain technology, which could enhance traceability, security, and trust concerns. Limitations: Issues related to distributed scrum agile software development specifically transparency, scalability, and lack of integration with blockchain technology persist.

[6] introduced a Communication and Coordination (C and C) framework grounded in the principles of Scrum methodology. This framework incorporates distributed agile teams, with a Scrum Master fulfilling the dual responsibilities of technical support and communication

coordination. The framework successfully achieves its objectives of communication, coordination, problem-solving, decision-making, and sprint backlog consensus, and it lacks the incorporation of blockchain technology to comprehensively address concerns related to traceability, security, trust, and transparency. Limitations: Issues related to distributed scrum agile software development specifically sprint backlog consensus persist and do not address distributed concerns related to traceability, security, trust, and transparency.

### Agile distributed software development

[17] This study investigates and ranks the challenges associated with scaling agile practices in a Distributed Software Development (DSD) environment. The Analytic Hierarchy Process (AHP) methodology was applied to prioritize these challenges and their respective categories based on their relative significance. This only provides analysis related to (DSD). This provides the analysis scaling agile in a distributed environment. The chainagile framework works in distributed decentralized scrum agile software development. Limitations: The limitations of this particular stuff are the challenges in the distributed scrum agile development environment of customer and developer related not addressed.

[18] This study offers a comprehensive framework aimed at establishing trust within distributed agile teams, addressing obstacles like limited face-to-face interactions, collaboration difficulties, time zone disparities, and cultural differences. Nonetheless, issues concerning security, traceability, and transparency persist. Limitations: Issues of distributed scrum agile development environment of customer and developer-related security, traceability, and transparency persist and are not addressed.

[1] This study introduces a hybrid agile software development and reuse approach known as software product line engineering based on Scrum. It integrates requirement engineering and design practices to establish a reference architecture for product management meetings. While this approach offers valuable insights and practical guidance for organizations aiming to enhance their software engineering practices by combining agile development and software reuse capabilities, it doesn't fully address challenges related to time zones and cultural differences. Moreover, concerns regarding security, traceability, and transparency persist. Limitations: Issues of distributed scrum agile development environment of customer and developer-related security, traceability, and transparency persist and are not addressed.

[19] The objective of this paper is to employ computational intelligence techniques to pinpoint and prioritize agile challenges in offshore software development, with a particular focus on reviewing communication and coordination challenges. Nonetheless, issues concerning security, traceability, and transparency persist. Limitations: Distributed scrum agile development environment security, traceability, and transparency of customer and developer-related persist.

[20] This study offers a software development framework that underscores the significance of agile methodologies, such as Scrum, in developing and enhancing Agile methods and practices in developing the software. No use of any technology which provides and controls the challenges of distributed environments. Limitations: Distributed scrum agile development environment security, traceability, and transparency of customer and developer-related persist.

These discussions collectively emphasize the ongoing need for an efficient framework to overcome significant challenges of traceability, security, transparency, remote location teamwork agreements, geographically dispersed teamwork efficiency, and trust within SADSD, leveraging the capabilities of blockchain technology. Existing frameworks have not fully addressed all these issues, particularly in scenarios involving customers and agile teams with

**Table 1. ChainAgile framework comparison with other related work.**

| Ref No | Related Work | Blockchain-based | Scalability | Communication | Coordination | Testing | Security | Prioritization | Review/Backlog |
|---|---|---|---|---|---|---|---|---|---|
| [21] | Social Networking tool Slack | N | N | Y | Y | N | N | N | N |
| [10] | Blockchain-Oriented Requirement Engineering | Y | N | N | N | N | Y | N | N |
| [12] | Blockchain Application for Agile | Y | N | N | N | Y | Y | N | N |
| [16] | Impact of Social media in distributed Agile | N | N | Y | Y | N | N | N | N |
| [22] | Software applications enable blockchain | Y | N | N | N | Y | Y | N | N |
| [18] | framework for building trust | N | N | Y | Y | Y | Y | N | N |
| [1] | Agile Distributed software development | N | N | Y | Y | Y | Y | N | N |
| [19] | Agile in offshore software development | N | N | Y | Y | Y | N | N | N |
| [13] | Fog Intelligence for Secure Smart Villages | Y | Y | N | N | Y | Y | N | N |
| [14] | Blockchain-based Authentication and Explainable AI | Y | Y | N | N | Y | Y | N | N |
| [15] | Blockchain and Deep Learning Empowered Secure Data Sharing Framework | Y | Y | N | N | Y | Y | N | N |
| **This Article** | ChainAgile Framework | Y | Y | Y | Y | Y | Y | Y | Y |

geographical differences. As a response to these needs, this study introduces the novel ChainAgile framework, aiming to bridge the gaps and limitations observed in previous frameworks and research endeavors. Our research introduces an innovative framework that combines blockchain technology to enhance Scrum Agile Distributed Software Development (SADSD). This framework effectively tackles trust-related challenges encountered by customers, clients, and development teams through the utilization of smart contracts for tasks such as acceptance testing, teamwork agreements, product backlog, task prioritization, remote teamwork effectiveness, secure payment processing, and the enforcement of penalties. Furthermore, ChainAgile provides a currency converter mechanism for converting ethers into fiat currency and successfully addresses scalability concerns by seamlessly integrating IPFS for the storage of JSON objects. This user-friendly approach significantly enhances the SADSD process. Table 1 depicts and summarizes the ChainAgile Framework Comparison with other Related Work.

## Materials and methods

In this section, we provide an overview of essential blockchain components utilized in the ChainAgile framework. These core elements include Decentralized Applications (DApps), the Ethereum Blockchain, the Interplanetary File System (IPFS), and Smart Contracts. Smart Contracts are embedded within the Ethereum blockchain programs. These are segments of code or programs. They function independently once specific predefined conditions are met [23]. These contracts, sometimes called transaction protocols, are scripted using the Solidity language, these smart contracts are executed on the blockchain to record events based on predefined terms and conditions [24].

The decentralized blockchain network of an open-source environment is Ethereum with the ability to execute smart contracts. It operates as a transparent ledger for recording and

authenticating transactions, accompanied by its native cryptocurrency, ether (ETH), for smooth transactions between network members. [25, 26]. Decentralized Applications (DApps) stand out as a prominent feature within Ethereum. DApps operate on a peer-to-peer network and they are digital programs or applications designed to, incorporate smart contracts equipped with user-defined code to carry out precise tasks. Interplanetary File System (IPFS) is a protocol that utilizes a peer-to-peer (P2P) network to store and distribute data in a content-addressable distributed file system [27]; [28]. IPFS ensures data integrity through cryptographic identifiers, preventing unauthorized alterations.

## Research methodology: Chainagile

The research methodology used for this research article is mentioned in Fig 3, how the research articles are searched and selected from different digital libraries. For this, we have chosen different articles based on titles, articles abstracts, frameworks, and models implemented specifically for distributed agile software development and search strings having keywords like blockchain, smart contract, agile software development, and distributed software development. Fig 4 shows the research article division according to the digital library. This shows how many research articles are found according to our search criteria and how many domain-related research articles we get from the digital library. Fig 5 shows the research article year-wise count, and how many research articles we get year-wise. Importantly, we have found very less articles on blockchain and its integration in distributed agile software development. The ChainAgile framework shows how the blockchain is integrated with distributed scrum agile software development and overcomes the challenges of a decentralized distributed environment by the use of blockchain. Table 2, shows the search strings using the keywords as per the research requirement to the digital search libraries.

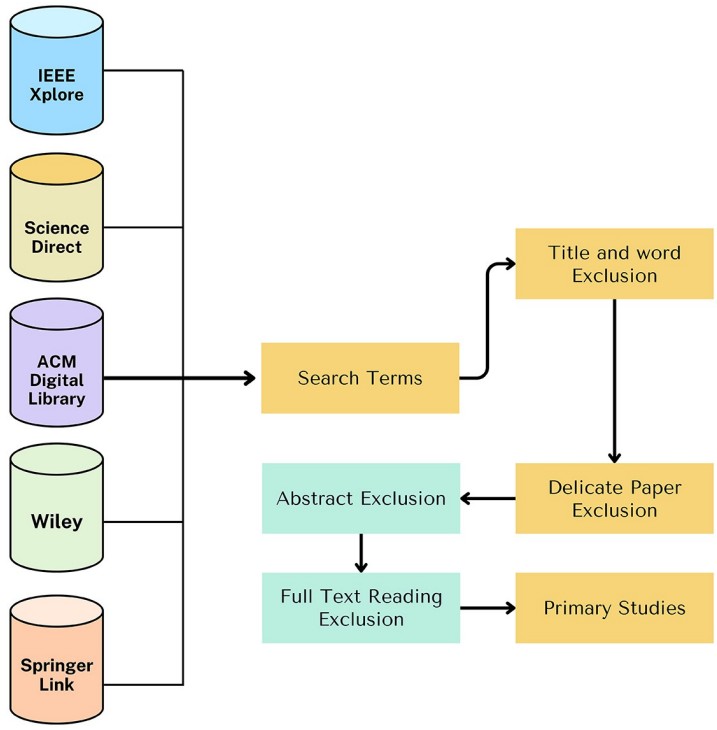

**Fig 3. Research methodolgy.**

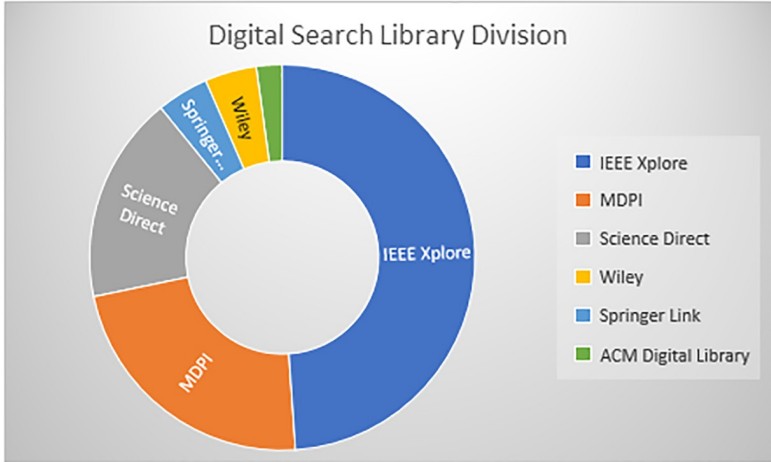

**Fig 4. Digital library research paper division.**

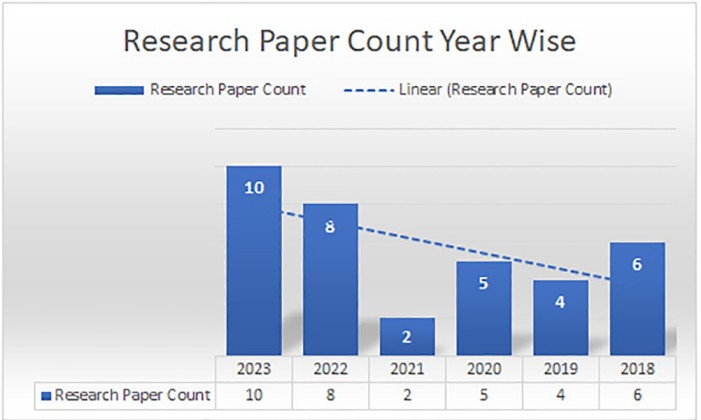

**Fig 5. Year wise research paper count.**

## The proposed framework: Chainagile

In this section, we introduce the ChainAgile framework, which integrates blockchain technology to manage Scrum Agile Distributed Software Development (SADSD) for clients and development teams spread across different.

The ChainAgile framework [29] utilizes a private Ethereum blockchain for efficient transactions. Blockchain technology is used as a central part of ChainAgile, encompassing activities such as customer feedback, task prioritization, product backlog, sprint backlog of distributed scrum agile, review of user stories, validation of developer payment through defined business logic, acceptance testing, and the secure transfer of payments to the digital wallets of development teams. The ChainAgile framework leverages blockchain capabilities to enforce accountability through automatic penalties for customers' payment delays or non-payments and developers' task completion delays. In the ChainAgile blockchain ecosystem, essential elements like infrastructure, Decentralized Applications (DApps), financial transactions, digital wallets, miners, consensus mechanisms, distributed ledger, privacy measures, and distributed

**Table 2. Search string.**

| Database (Digital Library) | Search Sting |
|---|---|
| | **Search String** |
| IEEEXplore | (("Blockchain" OR"Blockchain Technology") AND ("Agile software development" OR"Distributed Agile software development" OR"Scrum agile development") OR ("Smart Contract" OR"ethereum") OR ("Interplanetary file system" OR"IPFS")) |
| ACM Digital library | (("Blockchain" OR"Blockchain Technology") AND ("Agile software development" OR"Distributed Agile software development" OR"Scrum agile development") OR ("Smart Contract" OR"ethereum") OR ("Interplanetary file system" OR"IPFS")) |
| ScienceDirect | (("Blockchain" OR"Blockchain Technology") AND ("Agile software development" OR"Distributed Agile software development" OR"Scrum agile development") OR ("Smart Contract" OR"ethereum") OR ("Interplanetary file system" OR"IPFS")) |
| SpringerLink | (("Blockchain" OR"Blockchain Technology") AND ("Agile software development" OR"Distributed Agile software development" OR"Scrum agile development") OR ("Smart Contract" OR"ethereum") OR ("Interplanetary file system" OR"IPFS")) |
| MDPI | (("Blockchain" OR"Blockchain Technology") AND ("Agile software development" OR"Distributed Agile software development" OR"Scrum agile development") OR ("Smart Contract" OR"ethereum") OR ("Interplanetary file system" OR"IPFS")) |
| Willey | (("Blockchain" OR"Blockchain Technology") AND ("Agile software development" OR"Distributed Agile software development" OR"Scrum agile development") OR ("Smart Contract" OR"ethereum") OR ("Interplanetary file system" OR"IPFS")) |

storage create a comprehensive solution for managing projects in the Distributed Scrum Agile Software Development (DSASD) domain. The framework emphasizes secure payment transactions and user data privacy, offering infrastructure for user-friendly DApps, and facilitating communication, and coordination. This empowers dispersed users, providing control over information and enhancing transparency in data management. ChainAgile comprises six distinct layers, as shown in Fig 3, and encourages customer participation across all layers to facilitate effective tracking of development team progress through blockchain traceability [30]; [31]. Fig 4 provides a complete overview of the proposed ChainAgile framework implementation. Smart contracts play an active role in the testing layer, payment layer, and prioritization layer, where they are employed for conducting acceptance testing, verifying the payment requisites of developers, and task allocation and prioritization respectively. These smart contracts also play a crucial role in enforcing all mutually agreed terms and conditions between customers. clients and developers. In cases where user stories are not completed within the specified deadlines, ChainAgile automatically imposes penalties on the developers. Similarly, if customers make late or non-payments, penalties are assigned to them as well. All these actions are carefully stored within the private Ethereum blockchain. In the ChainAgile framework, all participants must complete a registration process by providing their credentials. This involves project managers and developers supplying information like their name, contact number, qualifications, skills, email, and experience. Simultaneously, customers desiring engagement with ChainAgile are required to complete a registration process, which requires the submission of their personal information including Name, Email, Project Domain, and Contact Number. The administrative team is entrusted with the task of initializing a new project for each customer, followed by the assignment of a dedicated project manager and expert developers responsible for executing project-related activities. During this process, all stakeholders receive a private key that serves as their gateway to the specific project. The key is particularly recorded within ChainAgile's blockchain database representing the agreement layer. This method stored the essential information related to customers, developers, communication messages, group chat logs, and video conferences. This strategic choice significantly boosts the

effectiveness of blockchain transactions by alleviating the load of data storage. [32]. Each piece of information stored in IPFS is linked to a unique IPFS hash, which is subsequently logged in the blockchain for future retrieval. However, it is imperative to note that data within the six-layer framework of ChainAgile continues to reside within the Ethereum blockchain. To efficiently handle extensive data volumes or storage requirements additional resources are introduced to the system. Within the branch of blockchain technology, the management of substantial transaction data within tight timeframes presents a challenging scalability. In response, ChainAgile incorporates the utilization of IPFS as distributed storage. IPFS leverages cryptographic hash functions and a peer-to-peer (P2P) network structure to facilitate the decentralized storage of data. Significantly burdening the blockchain with an excessive volume of user records, ChainAgile adopts the practice of storing such records on IPFS. Furthermore, ChainAgile extends this approach to encompass the storage of all messages, group chat logs, and records of video conference meetings on IPFS. By offloading this substantial data volume from the blockchain, ChainAgile effectively encourages transaction performance, resulting in faster and more efficient operations.

The role of Blockchain in distributed scrum agile software development using smart contracts plays a pivotal role in securing and validating digital transactions by utilizing a decentralized and distributed ledger. It ensures transparency, as each participant in the network has access to the entire chain of transactions. The blockchain's importance lies in its ability to establish trust without the need for intermediaries, this technology offers secure, transparent, and efficient processes. Its decentralized nature enhances security, reduces fraud, and increases accountability, making it a transformative force in digital transactions and data management. The blockchain with smart contracts controls all the processes of distributed scrum agile software development. All transactions related to requirements, prioritization, sprint backlog, user story implementation, its design and development, testing, after deploy verification and after verification payments. If the developed task late due to some reason its penalty mechanism is implemented in the form of late payments or penalties from client side, from distributed development teams and from customer. For this Fig 7 explains the proposed framework for ChainAgile which integrates the blockchain with distributed scrum agile software development. The designed blockchain architecture of ChainAgile shows in Fig 8 and the blockchain layered structure shows in Fig 9. Blockchain layered explains how the blockchain works in all layers of distributed scrum agile software development. The precise role where IPFS (Interplanetary File System) is employed involves serving as an off-chain storage solution within the ChainAgile framework. IPFS is strategically integrated to store pertinent data related to registered customers, developers, communication messages, group chat logs, and video conferences. This inclusion aims to enhance the efficiency of blockchain transactions by alleviating the data burden on the blockchain itself. IPFS utilizes a peer-to-peer network for the storage and distribution of data, ensuring protection against unauthorized alterations and addressing scalability challenges in the blockchain. In a distributed and decentralized system, the combination of blockchain and the Interplanetary File System (IPFS) proves highly effective. Blockchain provides a secure and transparent ledger, ensuring the integrity of transactions. IPFS, as a decentralized file system, complements this by offering efficient and secure storage and retrieval of data. The pairing of blockchain and IPFS enhances data distribution, reduces redundancy, and ensures data availability across the network.

**Blockchain integration understandings.**   The proposed framework leverages cutting-edge blockchain technology to establish a transparent system for scrum agile distributed software development users. By integrating blockchain, the system ensures transparency, enabling all stakeholders, including customers, clients, and developers, to seamlessly trace and track the various stages of the development processes.

The proposed secure payment method incorporates smart contracts, ensuring the integrity of various transaction details such as requirements, agreements, prioritization, user story implementation, testing, and sprint deployment and payments. All these transaction records are immutably stored in the blockchain, providing a secure and transparent environment. The integration of blockchain with the scrum agile distributed framework ensures the integrity and transparency of transactions, eliminating the risk of modification, deletion, or tampering. The system operates without a single point of failure or centralized authority. Fig 7 presents how blockchain technology is integrated with scrum agile distributed framework and Fig 8 shows the blockchain architecture implementation layer-wise. Fig 9 shows ChainAgile blockchain layered structure implementation. Fig 10 shows the scrum agile distributed software development layer-wise process flow in a more innovative way.

The key stakeholders in the proposed framework include customers, clients, and the development team. At the core of the framework is a decentralized application (DApp), relying on digital storage, IPFS, blockchain network, smart contracts, and a web platform for user interactions. All users involved in the traceability system are linked through the DApp for accessing the blockchain network. Registration is mandatory for all participants to join and engage with the system.

Upon registration, each user is assigned a unique public address, serving as their identification across the network. IPFS is seamlessly integrated to facilitate data transmission between digital storage and DApp components. The system accommodates various user levels, with distinct roles managing, controlling, and authorizing processes based on their respective authorities. The DApp encompasses an initial coin offering (ICO), an ICO exchange, and the integration of third-party payment gateways, including digital currencies and local currencies. IPFS technology ensures the secure transfer of information among distributed networks and digital storage, overseen by end-users through their public addresses. All entries are meticulously recorded in the blockchain ledger, ensuring complete traceability. The system incorporates an ICO exchange to convert tokens into fiat currency, with client payments to the development team deposited into their associated digital wallets. The proposed blockchain is hybrid, preserving data privacy for all stakeholders while remaining accessible to specific authorities for authentication, auditing, and transaction verification.

Users access the DApp through the management system, utilizing their public addresses to manage wallets and profiles. The backend of the platform operates a continuous ICO and currency exchange. Smart contracts are pivotal in managing authentication and administration policies within the scrum agile distributed environment. They govern ICO and exchange connections and dynamically set rates for various currencies.

## A. The layered architecture of ChainAgile

ChainAgile follows a seven-layered blockchain architecture, which is based on the blockchain architectural style. The architecture is presented in Fig 5 and includes the following layers:

1) **INTERFACE LAYER** This layer encompasses the user-friendly application of ChainAgile, decentralized applications (DApps), and connecting the customers, clients, and developers using a web portal that serves as the gateway to the ChainAgile system. Within this layer, individuals equipped with digital wallets within the ChainAgile environment interact with the interface to originate the scrum agile distributed software development processes.

2) **APPLICATION LAYER** This layer contains transaction details, payment information, and different records within ChainAgile. These records encompass prototypes, design, and project agreements created between customers and development teams. Moreover, this

layer serves as the instrument for utilizing the ether cryptocurrency, bridging the gap between the interface layer and the fundamental business logic, represented through smart contracts.

3) **BUSINESS LOGIC LAYER** This layer comprises the smart contracts operating within ChainAgile, responsible for managing terms and conditions. This layer serves as a dynamic repository for smart contracts. The layer facilitates the initiation, execution, and enforcement of contractual agreements and communication protocols.

4) **TRUST LAYER** The layer plays a pivotal role in supervising consensus algorithms, encompassing proof-of-work and Byzantine fault tolerance. It is also responsible for conducting security assessments of smart contracts, validating newly introduced blocks, and establishing consensus protocols for transactions [9]. The results of these operations are stored within the blockchain layer.

5) **TRANSACTION LAYER** The transactions performed by users or smart contracts in ChainAgile manage operations like validation, block validation, mining, and transactions.

6) **INFRASTRUCTURE LAYER** This layer provides a peer-to-peer network for distributing, validating, and dispatching transactions onto the Ethereum blockchain in ChainAgile. This network integrates verification, communication, and distributed networking mechanisms. Transactions are shared across the entire network, with each node carefully examining them based on predefined criteria. Upon validation, these transactions are securely added to the blockchain.

7) **SECURITY LAYER** The security layer holds a crucial responsibility in ensuring the safety of the blockchain network from various security threats [33]. It consists of a set of security algorithms and protocols that work in the ChainAgile system, carefully preserving its security. Users engage with the interface layer through a web portal and DApp, enabling them to perform actions like posting, commenting, chatting, and conducting project-related transactions using ether cryptocurrency. Development teams have the option to utilize a currency conversion service to transform ethers into traditional fiat currency. The transaction protocols are predefined by smart contracts [34], which encapsulate the core business logic within ChainAgile. The trust layer seamlessly integrates consensus algorithms to maintain the blockchain's integrity. ChainAgile employs the Ethereum blockchain implemented with a proof-of-work mechanism to maintain node authorization. The security layer seamlessly integrates with all the other layers to guarantee authorized access for participating nodes.

## B. ChainAgile layered structure

The ChainAgile framework is structured into six distinct layers, as depicted in Fig 6. Transactions within each of these layers are meticulously documented within the blockchain, ensuring that all members of the blockchain network are kept informed.

**Agreement: Layer 1.** Customers, clients, and developers collaboratively define their specific terms and conditions related to (SADSD), which is converted to smart contracts within the blockchain. These smart contracts are essential prerequisites for completing acceptance tests and activating payment options within the ChainAgile framework. The agreement layer serves as the platform for achieving consensus between customers, clients, and the development team to establish terms and conditions. This consensus is formalized through smart contracts, which are securely stored within the blockchain network, facilitating subsequent

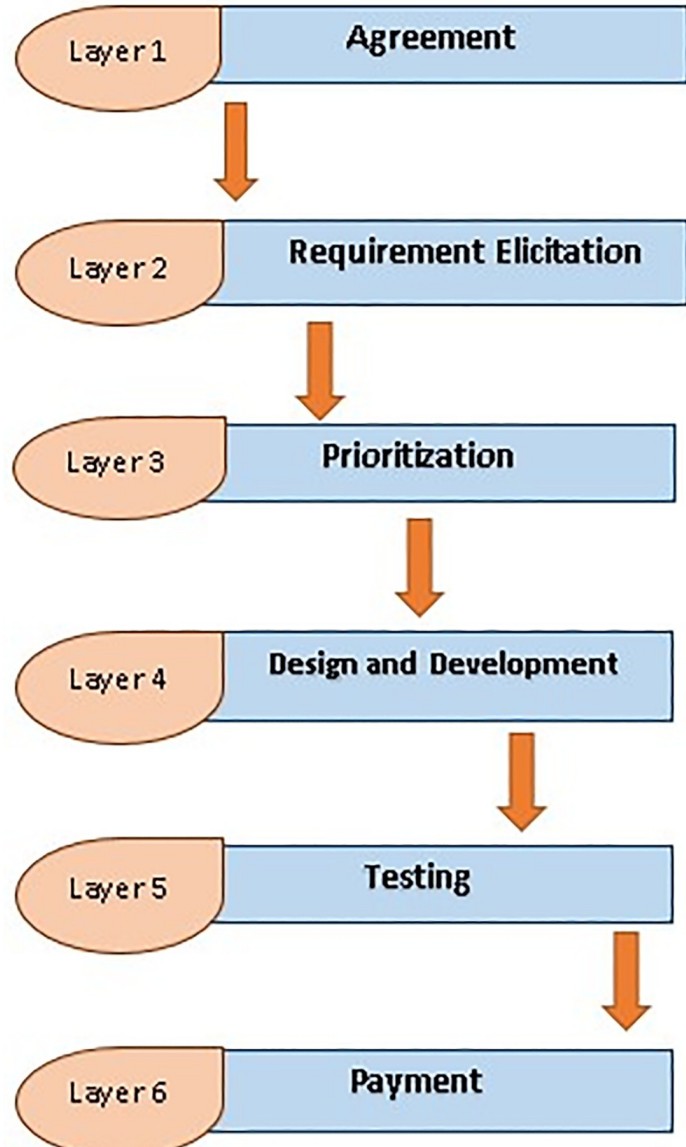

**Fig 6. Layers in ChainAgile framework.**

activities in the requirement elicitation layer. Once the agreement is set, the project's status transitions to the "Requirement Elicitation" phase, and this pivotal information is seamlessly updated within the blockchain network [35]. ChainAgile framework forces the Ethereum blockchain to deploy smart contracts. These smart contracts encompass transaction data and code responsible for executing transaction logic among stakeholders [36]. A fundamental aspect of the blockchain's operation is its immutability principle, ensuring that once a smart contract is deployed, its code remains unchanged. In situations involving bugs or errors, developers must create a new smart contract, rectify the issues, and deploy the revised version onto the blockchain. This process effectively transfers all existing data to the new smart contract while maintaining data integrity and strengthening security measures. The ChainAgile framework employs smart contracts as its core mechanism. The deployment process is recorded as a

transaction and becomes a permanent entry in the blockchain. A comprehensive overview of the ChainAgile framework's operational processes ChainAgile layers wise is depicted in Fig 7. The contractual details presented by customers, clients, and developers, implemented on the Ethereum platform in the form of smart contracts, are organized as JSON objects. These smart contracts confirmed all predefined terms and conditions were mutually completed in the agreement layer.

**Requirement elicitation: Layer 2.** ChainAgile framework, customers can initiate the requirements-gathering process by creating user stories, which are visualized on a virtual wall. This innovative framework encompasses five distinct communication modes such as messages, comments, video conferencing, group chat, post, and smart contract deployment process illustrated in Fig 9. These user stories, representing crucial project requirements, are thoroughly recorded as transactions within the blockchain. The development team actively engages with customers by responding to their posted user stories and, when necessary, scheduling or interacting with any communication mode for more in-depth discussions. Particularly, all posts and comments generated during these interactions are securely stored within the blockchain. Simultaneously, on IPFS group chats, messages, and video conferences are hosted, enhancing the overall efficiency and organization of communication. Once agreement is reached between the customer and the development team for the user stories, the backlog list is approved and recorded as a smart contract, marking its status as "Prioritization" and proceeding to the prioritization layer. This transition is securely logged in the blockchain, ensuring immediate notifications to all network members. Furthermore, the customer maintains the flexibility to implement changes or introduce additional user stories in the project. Importantly, these modifications are instantly communicated to all participants within the

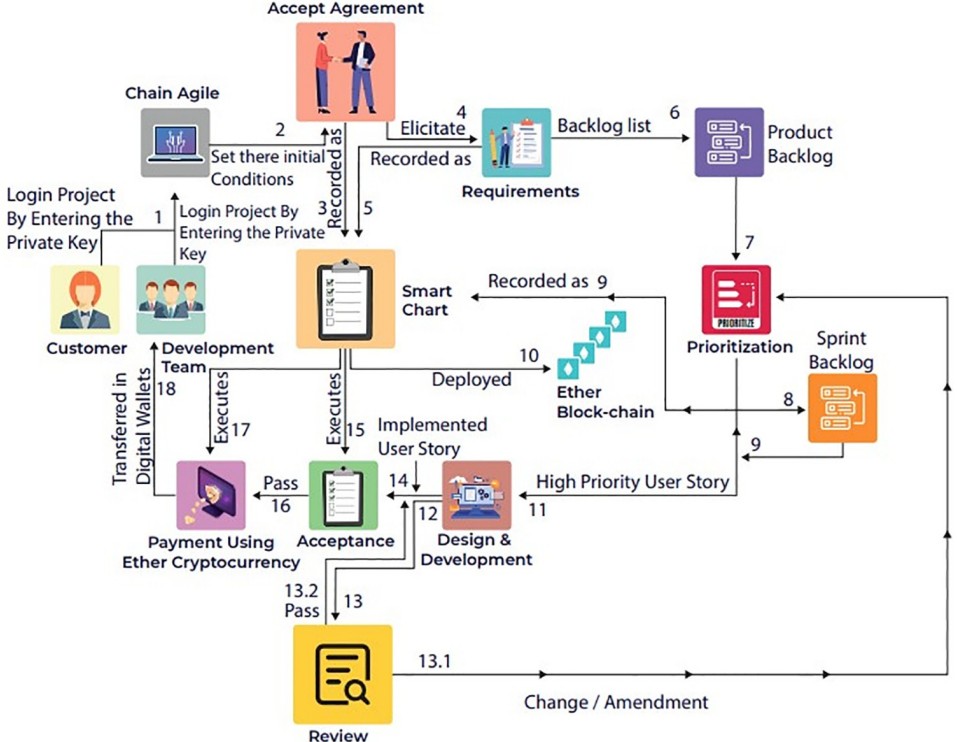

**Fig 7. ChainAgile framework.**

blockchain network, promoting transparency and informed decision-making throughout the project's lifecycle.

**1.JSON objects for the agreement of customer: agreement layer**

```
Customer's Agreement
(
"Project budget": "n decided by the customer"
"Project deadline": "n weeks decided by the customer"
"Each user story's deadline": "n weeks decided by the stakeholders
through consensus"
"Requirements": "Project must be completed within time and budget"
)
```

**2.JSON objects of agreement layer for the developer's agreement: agreement layer**

```
Developer's Agreement
(
"Each user story's price": "n decided by all the stakeholders through
consensus"
"Payment deadline": "n weeks for each tested user story decided by the
development team"
"Requirements": "Customer must pay for the completed and tested user
story within the due date"
"Project Manager": "n"
"Software Developer": "n"
"Tester": "n"
"UX Designer": "n"
"UI Designer": "n"
)
```

**Prioritization: Layer 3.**   In ChainAgile, the process of prioritizing user stories is a collaborative effort between the customer, client, and the development team within the prioritization layer. This collaboration may involve various communication methods such as comments, posts, video conferences, or group chats, all aimed at identifying the most essential user story to be prioritized first. Once a user story is successfully prioritized, its status changes to "Design and Development," signifying its readiness for the next phase. The high-priority user story is then seamlessly moved to the development layer. The prioritization process successfully defined the user story at this stage the sprint backlog and product backlog are also maintained which completes the prioritization process before storing all user stories. Importantly, this transition is particularly recorded within the blockchain network, ensuring that all participants in the blockchain network are promptly informed of this crucial development.

**Design and development: Layer 4.**   In the design and development layer of the ChainAgile framework, the development team takes responsibility for crafting prototypes for high-priority user stories. These prototypes are subsequently shared on the implicit block for customer review and feedback. Customers engage with these prototypes by offering reviews [33]. Additionally, both parties have the option to schedule video conferences to facilitate more in-depth discussions and exchange of ideas and solutions. In cases where the customer provides positive feedback, the development team proceeds with the implementation of the specific user story. However, if the feedback is negative, the team initiates a redesign of the prototypes to better align with the customer's requirements. Upon successful implementation of the user story by the development team, its status is updated to "Implemented" and securely stored within the blockchain. This transition marks the beginning of the testing phase for the implemented user story, and its status changes to "Testing." Importantly, this change is also recorded within the blockchain network, ensuring that all network members are duly informed of this progression.

**Testing Layer: 5.** The ChainAgile framework integrates an important testing layer aimed at validating the successful implementation of each user story. In the validation process for acceptance testing, specifically designed smart contracts are executed. These smart contracts check and verify whether the terms and conditions agreed between the customer and development are met or not. When all conditions are successfully satisfied, the test is marked as a "Pass," signaling the user story's progress to the payment layer. Importantly, every step in this process is carefully recorded within the blockchain network, ensuring that all network members are kept informed. However, in cases where the acceptance test fails due to missing requirements or other issues, the development team is promptly notified, and the user story is subsequently returned to the design and development layer for further refinement. The user story status is distinctly labeled as "Fail" and securely stored within the blockchain network. ChainAgile implements an acceptance test status of "Pass/Fail,". Additionally, the smart contract will automatically designate the user story's status as "Overdue" and enforce a related penalty, If the development team fails to pass the test within the timeframe. All alterations in status and penalties related to overdue user stories are diligently recorded within the blockchain network, thereby ensuring that all network members remain well-informed. The results of the acceptance test, illustrated in JSON objects conclusively confirm whether the terms and conditions have been met properly in the agreement layer.

**Payment Layer: 6.** By utilizing a private blockchain, ChainAgile introduces an additional layer of security that ensures transaction confidentiality and restricts node involvement in deploying consensus protocols and accessing transaction records. To enhance security the integration of private blockchain technology serves as a fundamental measure of the 51 percent attack mechanism within the ChainAgile framework.

**3.Acceptance test JSON objects in the testing layer**

```
Acceptance Test 1 (Pass State)
(
"User story ID": "01"
"Requirements": "All the terms and conditions of the customer in
agreement and backlog list must be achieved"
"Acceptance test date": "n date"
"Acceptance test status": "Pass"
)
```

**4.Acceptance test JSON objects 2 in the testing layer**

```
Acceptance Test 2 (Fail State)
(
"User story ID": "02"
"Requirements": "All the terms and conditions of the customer in
agreement and backlog list must be achieved"
"Acceptance test date": "n date"
"Acceptance test status": "Fail"
)
```

As the acceptance test is successfully passed by the high-priority user story, the ChainAgile framework seamlessly initiates the payment layer. Within this layer, an invoice is automatically generated for the completed user story, and a payment request is forwarded to the customer. The payment process requires the customer to make the payment by transferring ethers to the development team's digital wallets within the ChainAgile platform. These ethers can subsequently be converted into various fiat currencies, through the utilization of a cryptocurrency conversion service.

The comprehension method of payment and ether conversion procedures within ChainAgile are explained in Fig 8. Additionally, JSON objects represent key data. These cover various aspects, including agreement acceptance, requirement prioritization, Sprint Backlog, Design

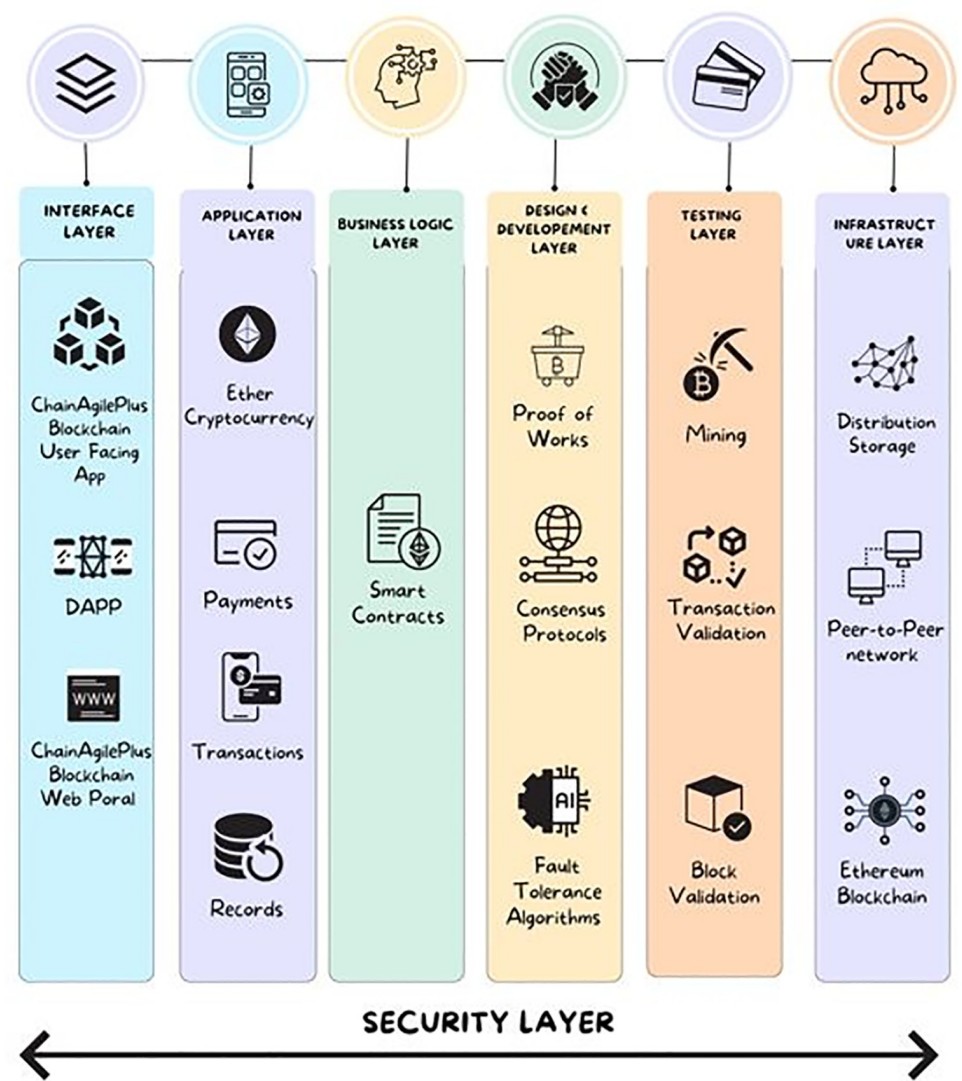

**Fig 8. ChainAgile blockchain architecture.**

and development, review, prioritization, and user story implementation, offering comprehensive insights into the ChainAgile framework.

**5.Acceptance agreement JSON objects in the agreement layer** Accept Agreement

```
(
"Private Key":"n"
"Login Project": Enter the n private key to the login project"
"Login Success": "If n is true"
"Login Fail": "If n is false"
"Initial Conditions":"n conditions set by all stakeholders consensus"
"Project Budget":"n decide by the customer"
"Project Deadline":"n time decided by the customer and development
team"
"Penalty": "For both sides if they don't fulfill the conditions."
"Accept Agreement": "Make a document with initial conditions and sign
by both sides."
```

```
"Record it": "Record the document as a smart chart"
"Elicitate Requirements": Elicitate all the user story for
requirements"
"Record it": "Record all requirements as smart chart"
)
```

**6.Requirement prioritization JSON objects in the prioritization layer**

```
Requirement Prioritization
(
"Developments Team": "n"
"Requirements": "Development team Elicitate through user story"
"Backlog list": "Maintain by the development team as product backlog
according to the requirements"
"Prioritization": "User set the priorities of product backlog"
)
```

**7.JSON objects for requirement Sprint Backlog in the Elicitation layer**

```
Requirement Sprint Backlog
(
"Prioritization": "User set the priorities of product backlog"
"Sprint Backlog": "Owned by development team according to user
prioritization"
"Development Team ":"n time decided by the development team for each
sprint"
"Record": "Record sprint as smart chart"
"Deploy": "Recorded data deploy to ether block-chain"
)
```

**8.JSON objects for Design and Development in the design layer**

```
Design and development
(
"Prioritization": "User set the priorities of product backlog"
"Design and Development": "Development team design and develop high
priorities user story."
"Developments": "n"
"Designers": "n"
"Review": "User review the design that is developed by the development
team"
"Changes": "If requirements are not full, customer address changes
according to priorities"
"Pass": "If requirements are full, the customer passes the review"
)
```

**9.JSON objects for review in the development and testing layer**

```
Review and Prioritization
(
"Input": "Design and Development"
"Review": "User review the design that is developed by the development
team"
"Changes": "If requirements are not full, customer address changes
according to priorities"
"Prioritization": "If any change is required in review, this require-
ment is sent to the Prioritization step"
"Pass": "If requirements are full, the customer passes the review"
"Implementation": "When the user passes review implementation start"
)
```

**10.JSON objects for user story implementation and acceptance in the testing layer**

```
User Story Implementation and Acceptance
(
"Input": "After review pass"
```

```
"Implementation": "Developer Team Implement the user story"
"Developer Team":"n"
"Acceptance": "When implementation is complete it is accepted"
"Execution": "All the smart chart record is executed"
"Payment": "After all execution user pays the payment to the Develop-
ment team"
"Penalty for Development Team": "If the development team does not
implement user story before or on deadline"
)
```

Fig 9 provides insights into the operational dynamics of the blockchain network within the ChainAgile ecosystem. ChainAgile ecosystem mentions the transaction, miners, privacy, distributed storage, infrastructure used in SADSD. This content provides an in-depth explanation of the core characteristics of blockchain technology. It defines blockchain as an immutable data storage system known for its decentralized, transparent, and peer-to-peer (P2P)

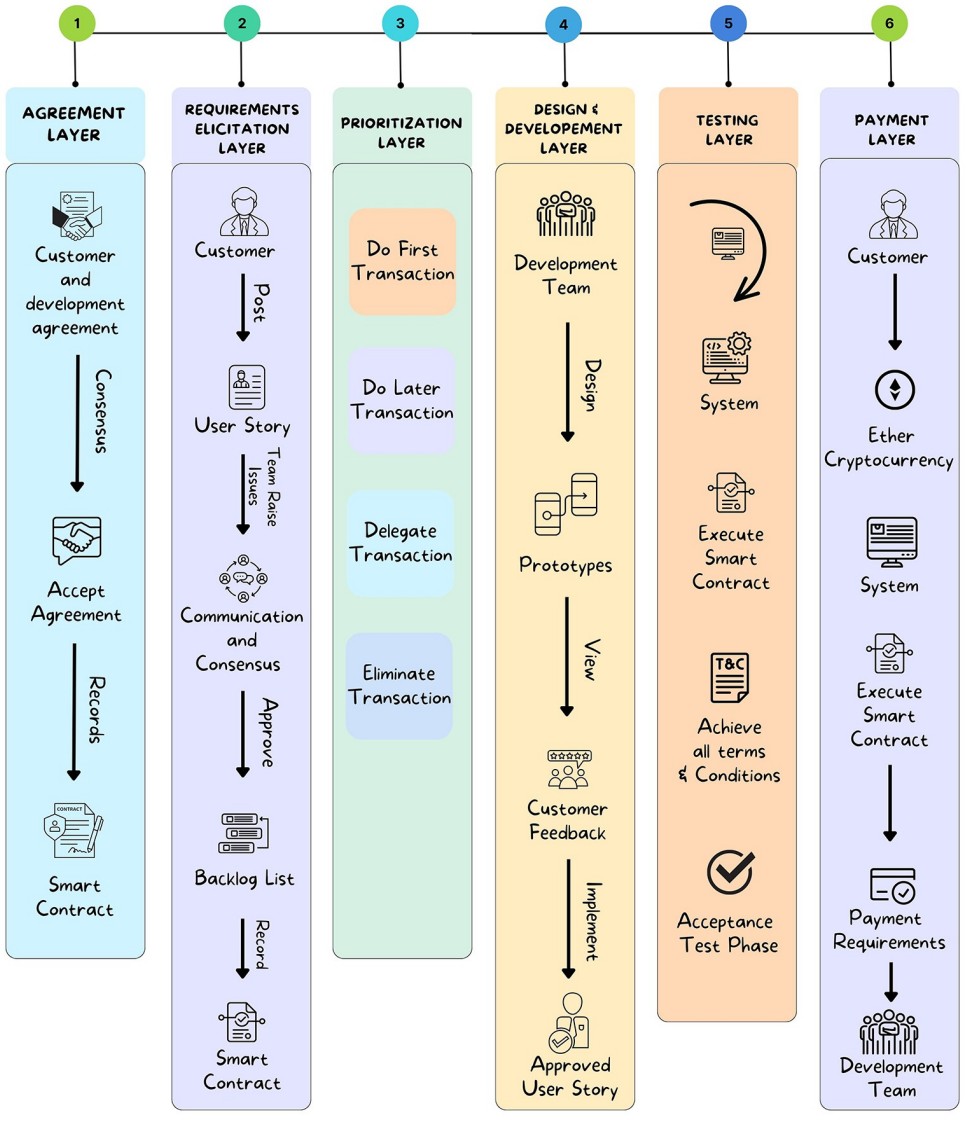

**Fig 9. ChainAgile layered structure.**

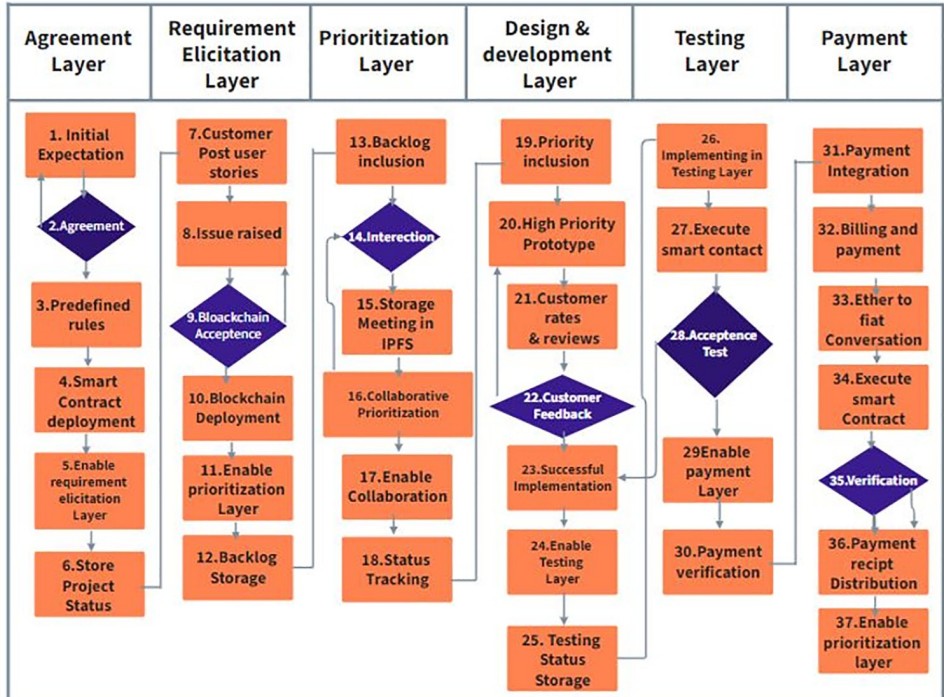

**Fig 10. Processes in layers of ChainAgile framework.**

architecture. In blockchain, data is permanently recorded in unchangeable data blocks. Blockchain working environment in the ChainAgile framework is shown in Fig 10. Penalties are imposed If the customer does not follow the defined payment requirements or submits payment after the designated deadline, through the smart contract's execution. If the customer fails to remit payment for the completed user story within a specific timeframe, their IP address is subject to being blocked, effectively preventing further access to ChainAgile.

Blockchain ledger formation explained in Fig 11, and how blockchain network works in ChainAgile framework explained in Fig 12. Additionally, Fig 13 offers a detailed illustration of the transaction process flow. The integration of the ChainAgile framework with blockchain infrastructure is a robust collaboration among its network participants, and users with various roles, including clients, stakeholders, developers, design engineers, requirement analysts, and blockchain engineers.

Detailed information regarding payment and penalties assigned to users is presented in JSON objects. Importantly, all transactions and penalties are securely stored within the blockchain, ensuring that all network members remain duly informed.

**11.JSON objects for the customer payment mechanism: payment layer**

```
Payment
(
"User story ID": "18"
"User story status": "Payment"
"Requirements": "All the terms and conditions of developers in the
agreement must be achieved"
"Payment due date": "n weeks decided by the development team through
consensus"
)
```

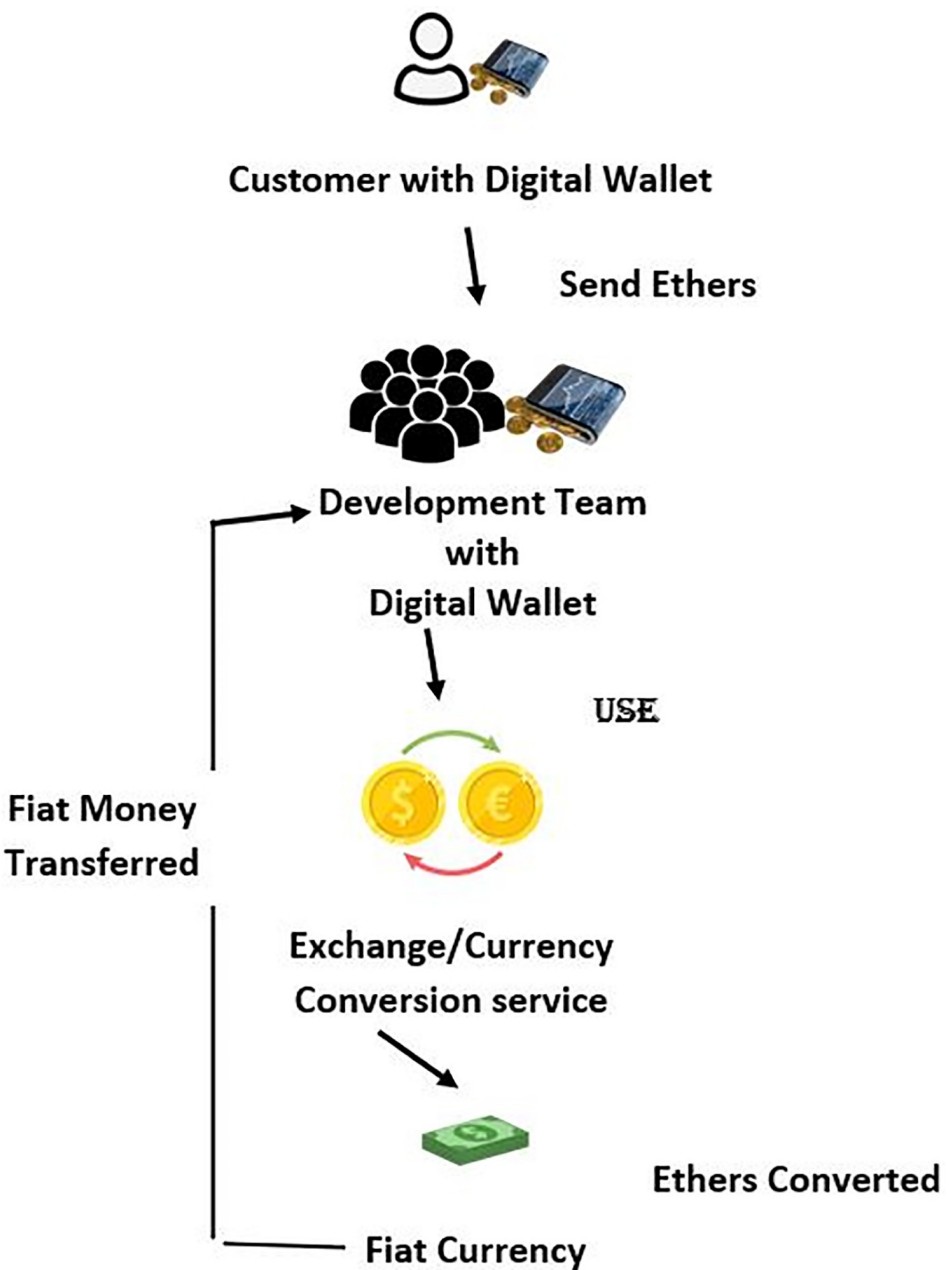

**Fig 11. Payment conversion in ChainAgile-framework.**

```
12.Penalty JSON objects for customers (related to late payment)
Late Payment Customer's Penalty
(
"User story ID": "04"
"Late payment deadline": "n weeks decided by the development team
through consensus"
"Payment status": "Late Payment"
"Penalty": "n decided by the development team through consensus"
)
```

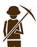

### Miners

Block chain miners add etherum transactions Data to the etherum Block chain in ChainAgilePlus. Every transaction is validated by miners in block chain for making it temper proof.

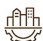

### Infrastructure

Blockchain provides infrastructure layers on which applications are build. In ChainAgilePlus framework, DApps are build on etherem blockchain.

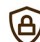

### Privacy & Identity

Customer and development team coordinate with each other without disclosing their identities. Data of every user in ChainAgilePlus is kept private by embedding blockchain technology.

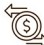

### Money Transaction

Blockchain deal with digital cryptocurrency for payments. In ChainAgilePlus, ethereum blockchain has been embedded that uses Ether as decentralized digital currency for money transactions.

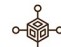

### Consensus Mechanism

Consensus is achieved in ChainAgilePlus when customer and developers agree with each other's initial conditions and accept agreement. Moreover, consensus is also achieved on backlog list in ChainAgilePlus

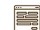

### Distributed Ledger

ChainAgilePlus embeds blockchain which provides multiple copies of data as it is a distributed electronic ledger. Multiple copies of data across the blockchain network ensures immutability.

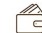

### DAPPS

DApps in ChainAgilePlus provide interface to the blockchain users. Customer and developer can connect to the ChainAgilePlus via DApps

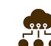

### Wallets

Customer and development team can send and receive Ethers in their digital wallets which are like bank accounts in ChainAgilePlus.

### Distributed Storage

Blockchain facilities ChainAgilePlus users with the decentralized storage which ensures that data is stored on multiple servers and solved latency issues. Therefore, no central system can control the data.

**Fig 12. ChainAgile ecosystem.**

13. **Penalty JSON objects for customers (related to no payment)**

```
Penalty for non-payment on Customer
(
"User story ID": "05"
"Late payment deadline": "n weeks decided by the development team
through consensus"
"Payment status": "No Payment within n weeks"
"Penalty": "IP address blocked"
)
```

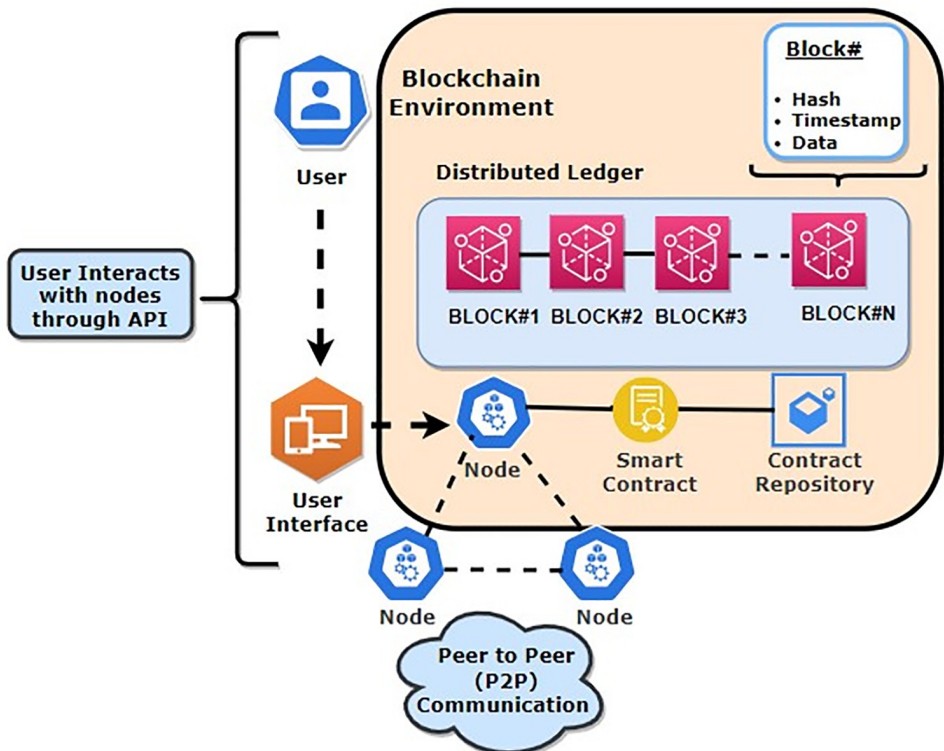

**Fig 13. Blockchain environment in ChainAgile framework.**

**14. Penalty JSON objects for developers within the payment layer**
```
(
"User story ID": "06"
"User story due date": "n weeks decided by the stakeholders through
consensus"
"User story status": "Overdue"
"Penalty": "n decided by the customer"
)
```

# Experiment and results

## Performance and evaluation

In this section, we showcase the real-time execution of the devised framework and delve into the outcomes achieved. The experiments operated under the following assumptions:

- No miner or coalition of miners possessed 51 percent hash power in the system.

- Access to data was exclusively granted to registered users.

  The functions included:

- GetChain for demonstrating the chain.

- ConnectNode for establishing connections between nodes.

- MineBlock for mining a block within the blockchain.

- addTransaction for involvement in blockchain transactions.

- chain-valid for validating the developed chain.

- replace-chain for replacing the chain with a newly entered one.

We assessed the efficiency of ChainAgile by implementing the blockchain network which creates data blocks on the network and analyzed the latency by getting the largest chain through the network. For this we utilized Python in this study, utilizing Replit as our integrated development environment (IDE). Replit, an online cloud-based service, operates as a Software as a Service (SaaS) for crafting smart contracts. Postman is employed for all HTTP requests used during the experiment engaged with the blockchain. Additionally, we leverage the Matplotlib library in Python to generate visual representations of blockchain size and data latency. The metrics utilized to evaluate in the ChainAgile framework include an analysis of latency and block performance.

**BLOCK SIZE** is the amount of data stored in a single block, usually containing transaction data. Evaluating block size is important to understand how quickly files or the blockchain typically grow.

**LATENCY**, in a system, refers to the delay experienced when one component waits for a response from another component after taking an action. In the context of a blockchain network, latency specifically represents the time gap between submitting a transaction to the network and receiving the initial confirmation of its acceptance by the network.

**Ethereum Cost/Gas** The proposed framework for scrum agile distributed software development involves stakeholders such as customers, clients, and development teams. It includes components like a decentralized application (DApp), which relies on digital storage, the Interplanetary File System (IPFS), a blockchain network, smart contracts, and a web platform for end-user interactions. IPFS is integrated to facilitate data transmission and storage of communication data, enhancing the overall functionality and connectivity for the involved stakeholders.

Ethereum, a prominent blockchain, records over one million daily transactions and utilizes smart contracts written in various programming languages. Smart contracts on Ethereum are self-executing programs that run when specific conditions are met, combining data and methods. The Ethereum Virtual Machine (EVM) compiles these contracts. Once deployed, the bytecode is stored on the blockchain, accessible via the smart contract's address. Transactions in ChainAgile require users to pay transaction fees in gas, a unit measuring computational costs. Miners are rewarded with Ether and Gas for their contributions to ChainAgile operations. Gas, akin to a transaction fee, compensates miners for executing smart contracts, with the actual price influenced by market conditions.

For example, certain operations in Ethereum entail specific gas requirements; for instance, the blockhash operation necessitates 20 gas units, and the ADD operation requires 3 gas units. By default, any operation impacting the state of the Ethereum Virtual Machine (EVM) requires a minimum of 21,000 gas. An illustration of this is the amount needed for transferring Ethers from one account to another.

In ChainAgile we have experimented and assessed the efficiency of ChainAgile by implementing the blockchain network which creates data blocks on the network and analyzed the latency by getting the largest chain through the network. Fig 13 shows the blockchain implementation environment in ChainAgile. Fig 14 shows the blockchain network. Fig 15 shows the ChainAgile transaction process flow. Fig 16 shows the experimental results of chain size of block. Figs 17–19 show the latency of the experiment. Fig 20 shows the block size and its increase in the experiment.

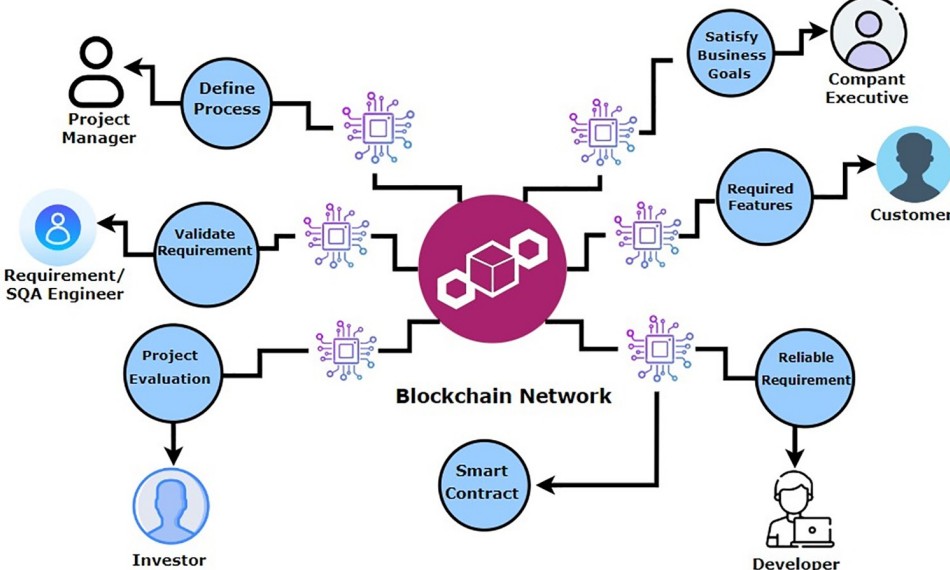

**Fig 14. Blockchain network working in ChainAgile framework.**

## Performance evaluation

In this section, we give a comprehensive exploration of the real-world performance evaluation of ChainAgile through practical experimentation. The ChainAgile blockchain network has been established to facilitate real-time data transfers and transactions. we configured the transactions in HTTP request forms using the Postman tool. To validate the effectiveness of our proposed framework, we conducted mechanisms that is, add-transaction, get-chain, replace-chain, connect-node, mine-block and is-chain-valid. In the Postman environment, we initiated a sequence of random transactions through HTTP requests, repeating this process by using the block size 900 times, 2000 times, and 5000 times repeatedly. This will clarify the results for performance evaluation more precisely. As the blockchain progressed, it displayed a size range from 0.284 KB to 750 KB over 900 blocks, 0.485 KB to 850 KB over 2000 blocks, and 0.590 KB to 990 KB over 5000 blocks. The introduction of new blocks with transactions resulted in an average size increase of 325 KB to 360 KB around for 900 blocks, 525 KB to 560 KB around for 2000 blocks, and 690 KB to 960 KB around for 5000 blocks, respectively. Fig 21 illustrates the relationship between the increasing number of blocks and the expanding blockchain size for 5000 blocks, it is big number in transactions. Fig 14 illustrates latency for adding 900 blocks, Fig 15 illustrates latency for adding 2000 blocks, Fig 16 illustrates latency for adding 5000 blocks respectively.

The incremental trend indicates a consistent rise in the blockchain size. The response time gradually increases as more nodes await in the pool for miners. The average chain size hovers around 0.04 MB to 0.4 MB for every 500 blocks in a 5000-block chain experiment. This size continues to grow with the addition of blocks to the chain. System latency is assessed by measuring the time taken to execute transaction requests, triggered at specific intervals to determine the system's latency rate. Consequently, the on-chain traffic is regulated throughout the process. Latency within the chain is influenced by the computational power of nodes, and adaptable consensus algorithms maintain blockchain efficiency during operations. The inclusion time of transactions in the blockchain remains constant, reducing

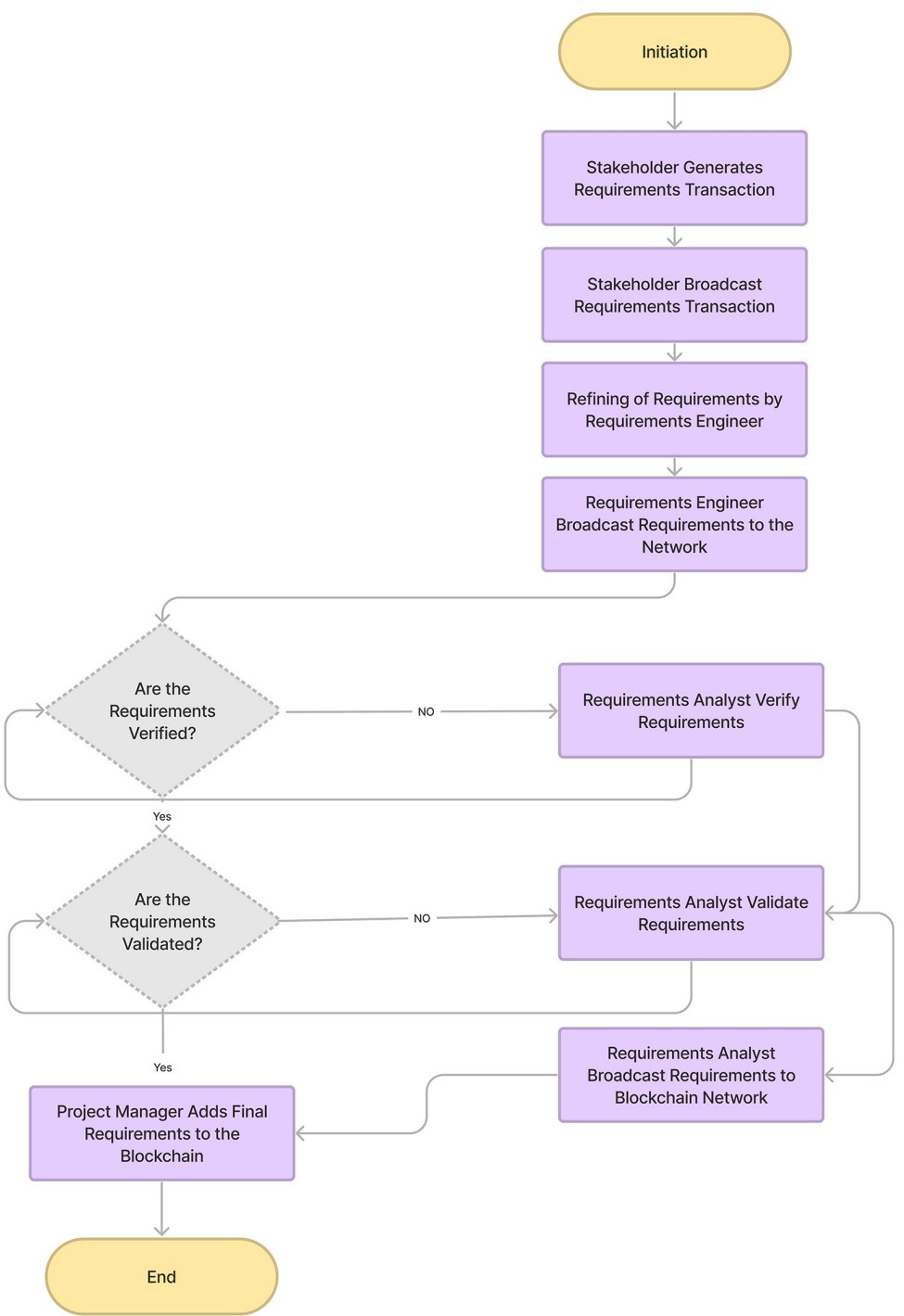

**Fig 15. Blockchain transaction flow in ChainAgile framework.**

system latency. However, the decentralized nature of the blockchain introduces some delays in data access.

Furthermore, the performance of these numerous servers within the ChainAgile blockchain network is susceptible to influences such as the response time of their local machines to

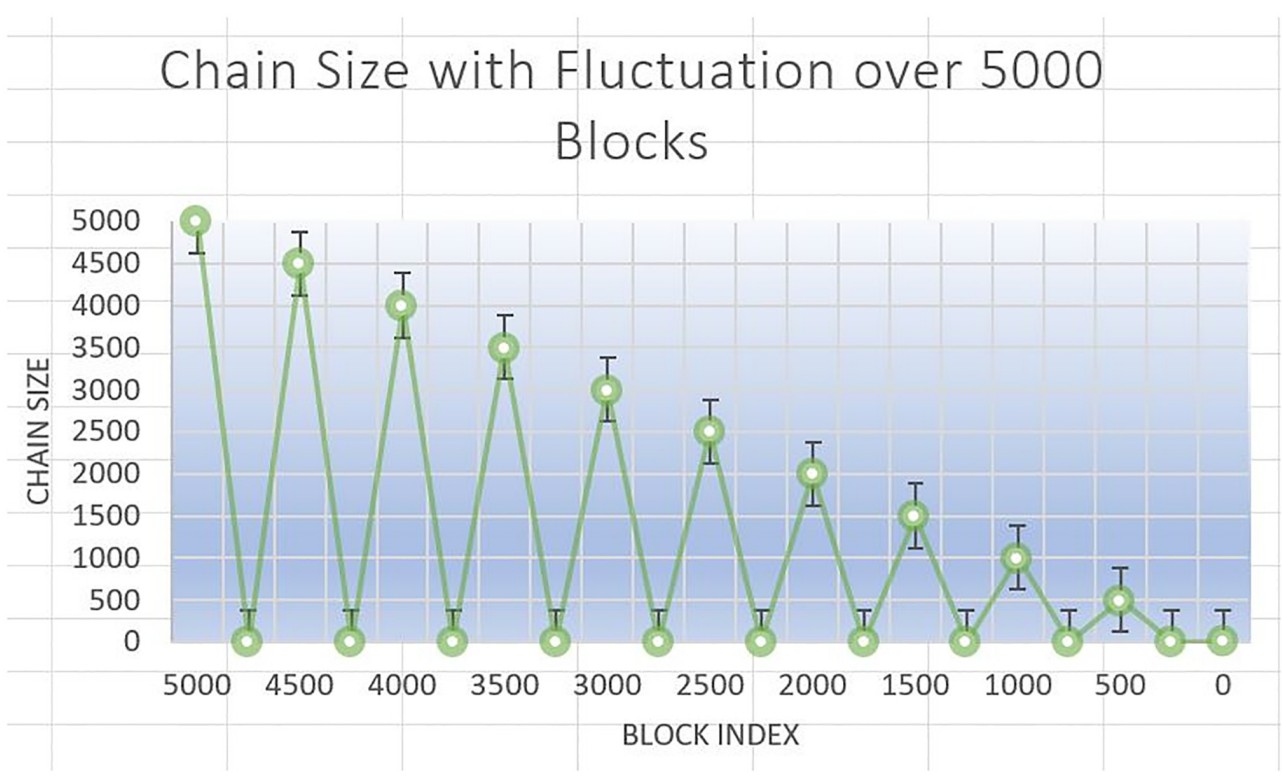

**Fig 16. The average growth in ChainAgile's file size.**

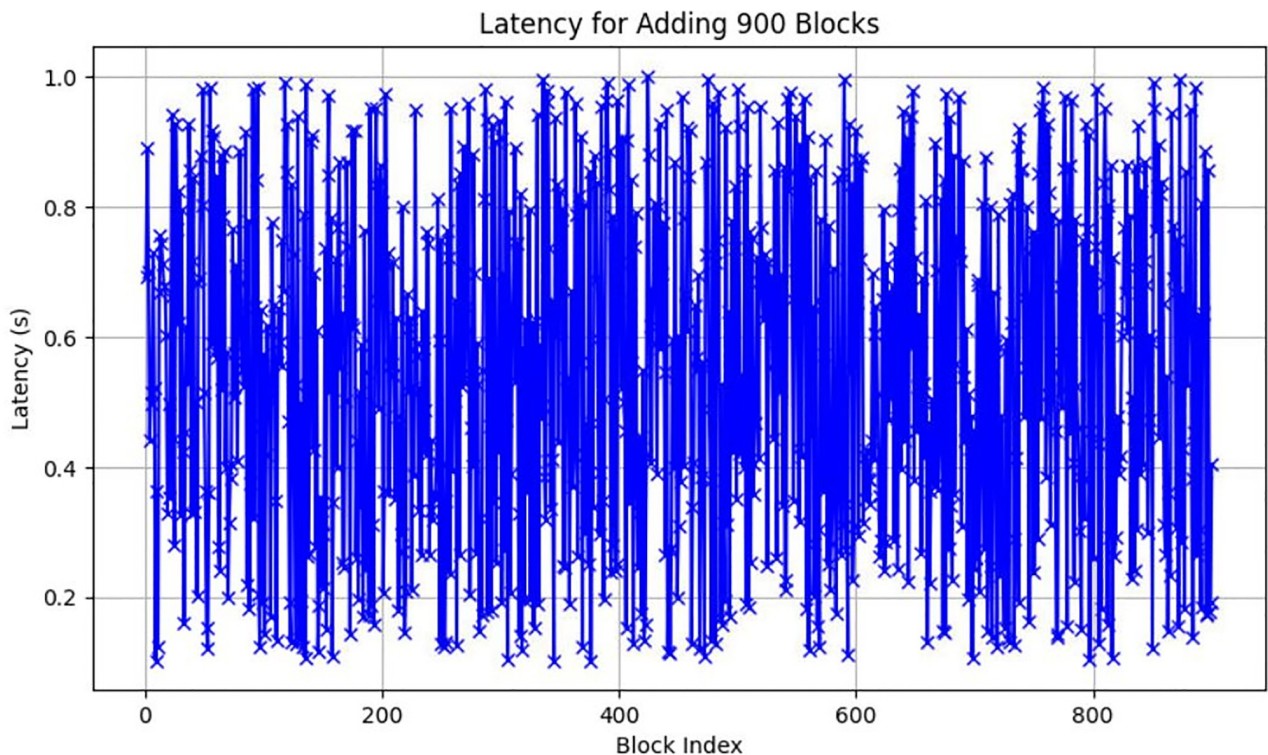

**Fig 17. Longest chain Latency in ChainAgile 900 blocks.**

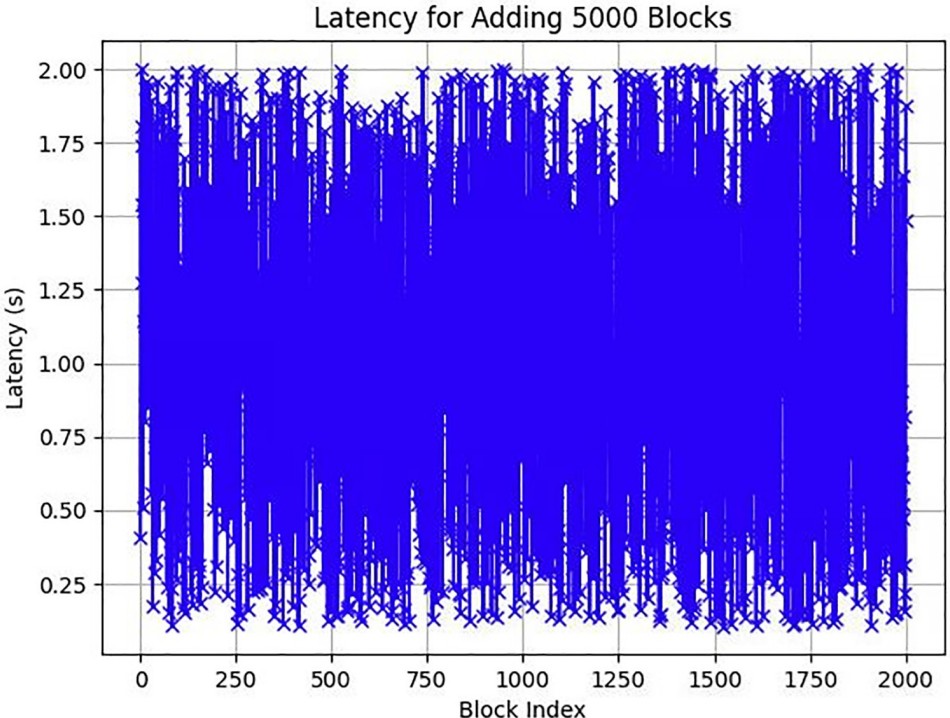

**Fig 18. Longest chain Latency in ChainAgile 2000 blocks.**

multiple requests, machine processing speed, and available internet bandwidth. These collective factors collectively contribute to the latency observed when executing HTTP requests across different servers.

**ChainAgile blockchain-based framework.** ChainAgile incorporates a private Ethereum blockchain to support fundamental principles such as traceability, trust, transparency, and security. ChainAgile reinforces security by mitigating the susceptibility to 51 percent of attacks. Ensuring secure payments and equitable distribution among virtual agile teams. Mandating the use of private keys for stakeholders to access specific projects, effectively restraining from unauthorized participation. ChainAgile actively promotes transparency within the software development process. It guarantees perpetual access to data for all geographically dispersed stakeholders, who receive timely notifications with every update. ChainAgile introduces trust through the execution of smart contracts, precisely designed to satisfy both customers and developers. These smart contracts serve as reliable mechanisms to verify the fulfillment of all instructed terms and conditions. ChainAgile precisely tracks the progress of work conducted by distributed agile teams, thoroughly documenting each step within the blockchain. This comprehensive information is disseminated to all network members, thereby ensuring complete traceability and accountability throughout the process.

In ChainAgile, we have integrated IPFS storage solution as an off-chain, effectively mitigating the scalability challenges often encountered in blockchain environments. IPFS serves as the repository for storing customer and developer records, along with their communication data. This strategic decision to offload such data from the blockchain substantially enhances transaction performance within ChainAgile, leading to expedited and more efficient operations.

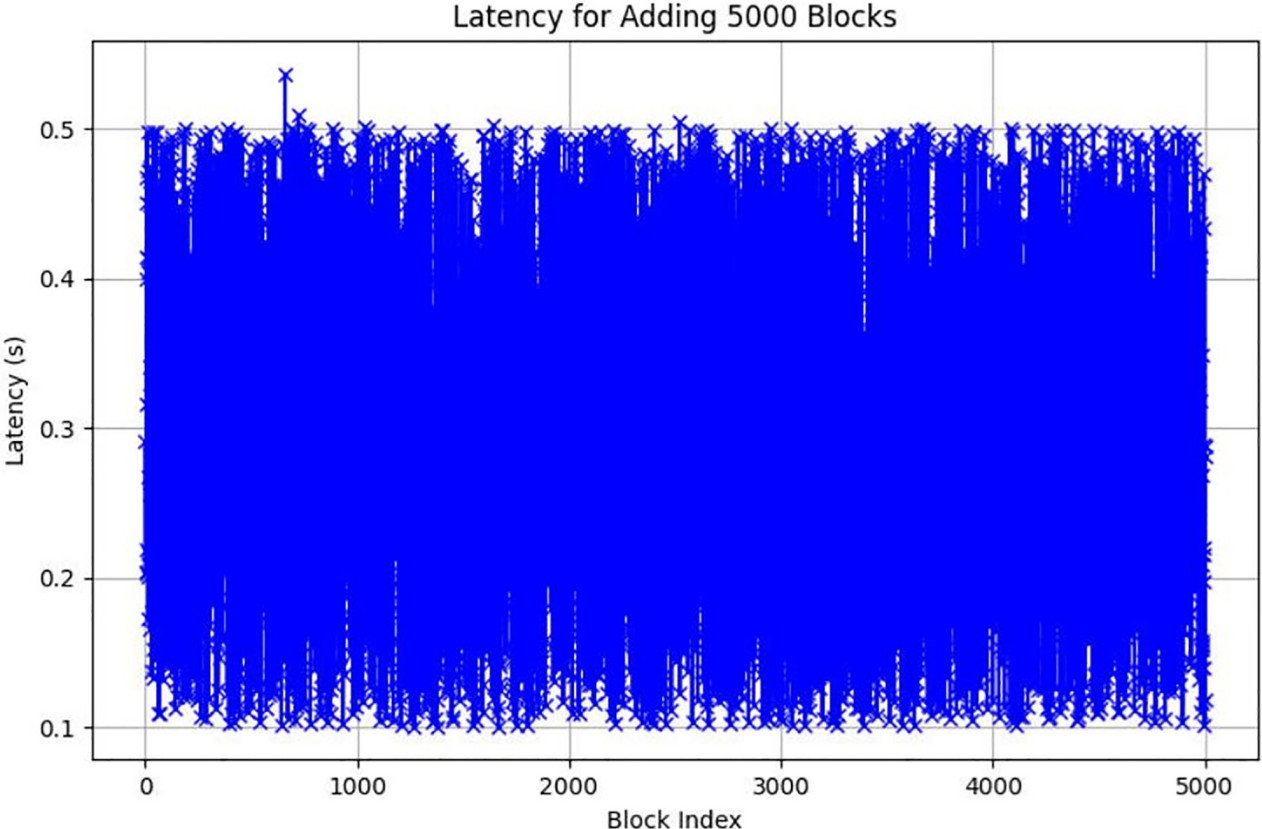

**Fig 19. Longest chain Latency in ChainAgile 5000 blocks.**

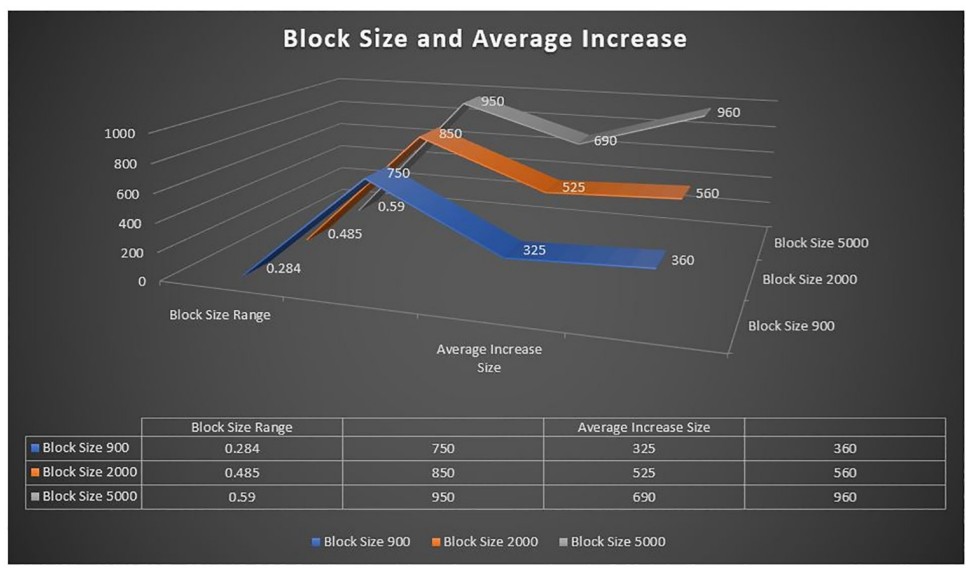

**Fig 20. Block size and average increase.**

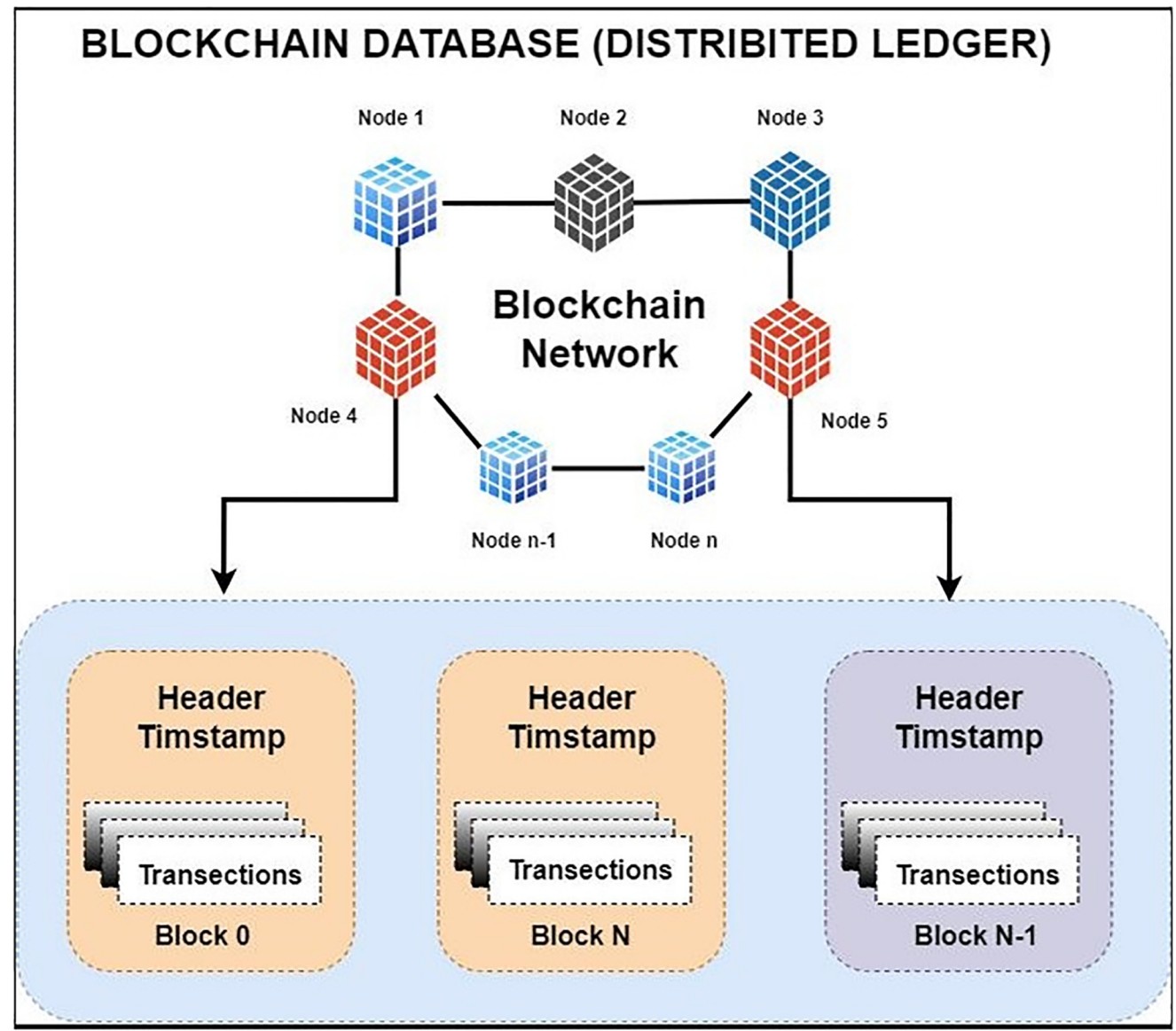

**Fig 21. Blockchain ledger formation in ChainAgile framework.**

ChainAgile utilizes a DApp (Decentralized Application) to provide a user-friendly interface. Within this interface, users are equipped with access to five distinct communication modes that span across all six layers. These modes encompass comments, group chat, posts, and video conferences, collectively delivering a comprehensive and highly effective communication experience.

In the proposed framework, consensus mechanisms have been integrated into the agreement and requirement elicitation layers. This integration serves the purpose of enhancing coordination between customers and developers.

ChainAgile connects the power of smart contracts for acceptance testing, thus ensuring the verification of the development team's devotion to all terms and conditions instructed by the customer.

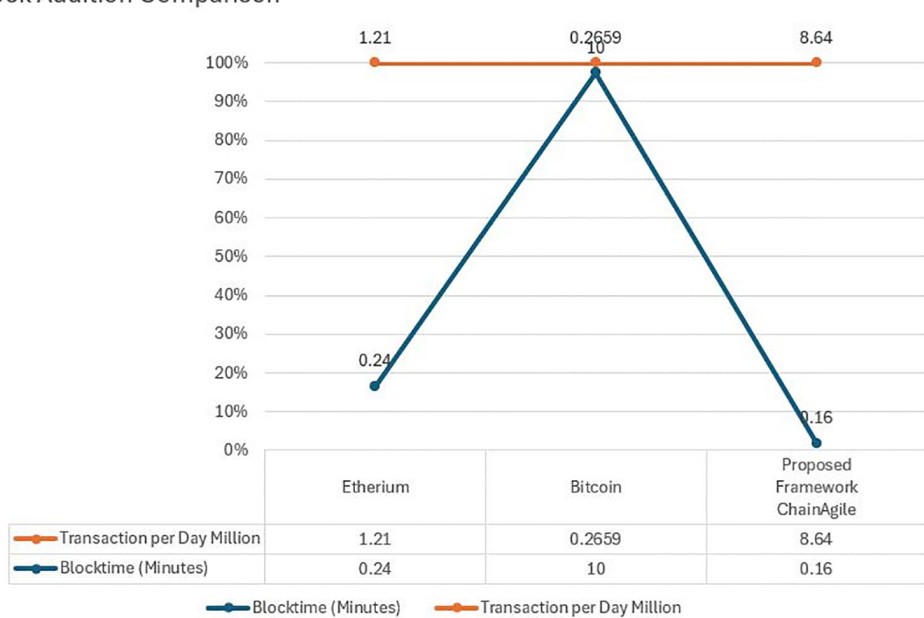

**Fig 22. Block addition.**

Fig 17 shows the representation of different block size chains 900, 2000, and 5000 blocks respectively. ChainAgile blocks average increase and average block size when a new block is inserted in the chain. This clarifies that at the time of start the increase and latency are slightly higher before it comes to average.

Ethereum Cost/Gas analysis for the proposed ChainAgile framework, ChainAgile CAT token value is designed with its initial cost of 1 USD with the token type ERC20 ethereum standard token for Ethereum blockchain network. A specific fee is charged for each transaction in ChainAgile, depending upon the currency exchange used ICO. The proposed framework demonstrates enhanced efficiency by adding new blocks to the blockchain more swiftly than existing networks. While Bitcoin takes up to 10 minutes and Ethereum 15 seconds, the proposed framework adopts a mechanism to add a new block approximately every 8.5 seconds.

Fig 22 illustrates the new block additions, highlighting the reliability and efficiency of our proposed system compared to others. The delay in blocks on Ethereum and Bitcoin networks results in 0.2659 million and 1.21 million transactions per day, respectively. In contrast, our research's proposed framework can add an average of up to 8.64 million transactions to the blockchain daily.

Moreover, our proposed framework exhibits faster confirmation of new transactions compared to Ethereum and Bitcoin. Notably, the average transaction fee in our framework is USD 0.5 initial and standard. Fig 23 shows the detail of cost/gas transaction fee.

## Discussion

This section presents a comprehensive discussion of the experimental results obtained from our ChainAgile framework. Our framework has been particularly designed for scalability, security, transparency, geographically dispersed teamwork agreements, geographically dispersed teamwork effectiveness and optimized performance. ChainAgile is a productive

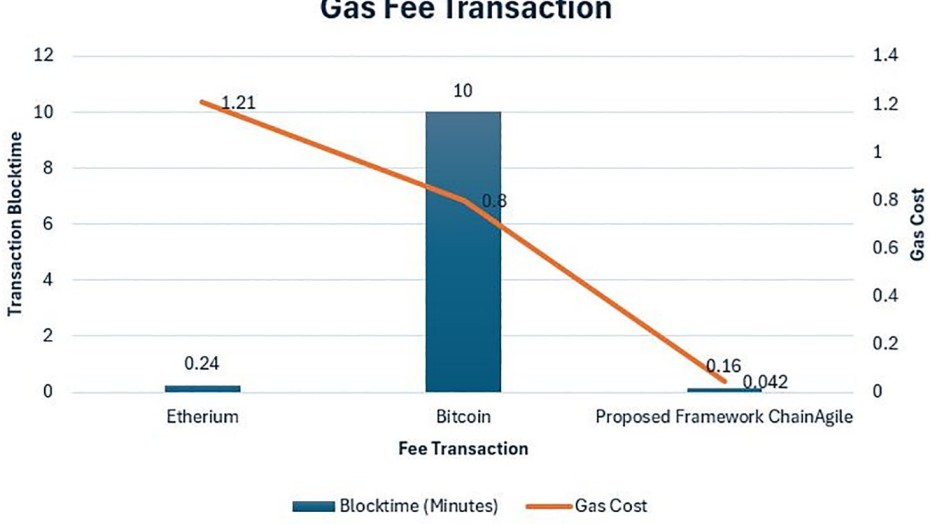

**Fig 23. Cost/Gas transaction fee.**

ecosystem designed to support the effective execution of Scrum Agile distributed software development-based projects, delivering a platform that encourages collaborative and geographically dispersed teamwork. In this framework, we proactively attempt the substantial challenges associated with Scrum Agile Distributed Software Development (SADSD). These encompass concerns related to trust, traceability, security, transparency, communication, geographically dispersed teamwork agreements, geographically dispersed teamwork effectiveness and coordination. By incorporating blockchain technology, we confront and overcome these challenges. The decentralized attributes, consensus mechanisms, distributed storage, and robust security inherent in blockchain play a pivotal role in mitigating issues such as project delays, payment disputes, project cancellations, customer dissatisfaction, and trust-related problems in a distributed environment. The performance evaluation of ChainAgile seamlessly incorporates blockchain, presenting users with six unique layers, each equipped with its workspace considerably customized for the distributed scrum agile software development life cycle. The advantages of blockchain integration into ChainAgile are: Blockchain's inherent decentralized nature seamlessly aligns with the requirements of SADSD. DApps provide a user-friendly interface for users of the distributed environment across six layers, enabling effective communication and collaboration among geographically dispersed users. Blockchain actively deploys consensus mechanisms to elevate coordination within the SADSD framework. The Ethereum blockchain integrated into ChainAgile automates critical processes such as acceptance testing, payment verification for developers, and seamless distribution of payments to the development team's digital wallets with smart contracts. Blockchain technology empowers users by providing digital wallets and cryptocurrency (e.g., ether, ETH) for efficient and secure payment transactions. This technology ensures a high level of transparency, ultimately leading to keen customer satisfaction. ChainAgile effectively mitigates the risk of 51 percent attacks, thus ensuring secure payment transactions. Our experimental findings emphasize the pivotal role played by blockchain technology as an initial element of this research, significantly elevating the software development process within the SADSD domain. Particularly, previous research endeavors did not propose a framework that combines efficiency, transparency, security, scalability, trust, and traceability, within SADSD. However, it's imperative to acknowledge

that the inclusion of the Ethereum blockchain within ChainAgile does come with certain limitations. These limitations encompass factors such as the substantial energy consumption during block mining and the inherent challenge of data modification once it has been securely stored on the blockchain. Furthermore, the results derived from our performance evaluation distinctly highlight the immense potential of blockchain technology within the SADSD. The blockchain infrastructure comprehensively encompasses all aspects of the scrum agile distributed software development [37] (Elicitation, Analysis, Validation, prioritization, product backlog, sprint backlog Management, Specification) processes. It acknowledges the existing challenges and will overcome gaps and limitations by implementing the innovative blockchain-driven model presented in this context effectively. Our empirical findings establish blockchain technology as a pivotal asset in this study SADSD. The integration of blockchain infrastructure holds promise for enhancing trust, sustaining security, and supporting reliability across these phases. The results of our performance evaluations indicate that blockchain technology possesses the capacity to transform the future of SADSD and has the potential to conduct substantial changes for offshore software companies.

## Conclusion

Existing tools and frameworks designed for SADSD lack fundamental features like security, transparency, traceability, geographically dispersed teamwork agreements, geographically dispersed teamwork effectiveness, and trust. This deficiency and challenges highlight the requirement for an efficient framework capable of addressing these significant challenges. In response, we propose an innovative solution called ChainAgile, which integrates blockchain technology to effectively confront the primary problems encountered in SADSD. ChainAgile utilizes DApps to provide an intuitive user interface, which records transaction and information in the blockchain. This storage mechanism allows for comprehensive tracking of work progress among geographically dispersed teams. The framework leverages smart contracts to automate several critical functions, including acceptance testing, payment verification, payment distribution, task prioritization, product backlog, sprint backlog and manage the penalties for late or non-payments. To ensure security against 51 percent attacks, we implement a private Ethereum blockchain, and to handle blockchain scalability concerns, we employ IPFS as off-chain storage. Our experimental findings conclusively demonstrate that ChainAgile, powered by blockchain technology, proficiently resolves the principal challenges faced by SADSD. In our future endeavors, we intend to enhance ChainAgile by incorporating features for user rating, distributed scrum of scrums, DevOps and Scrum integration with blockchain and reviews management. Moreover, we plan to integrate a translator to support multiple languages, further expanding the framework's accessibility and usability in other distributed environments of scrum agile software development. Additionally, we aim to employ smart contracts to address task distribution complexities among development teams operating within the ChainAgile framework.

## Supporting information

**S1 Dataset.**
(CSV)

## Author Contributions

**Conceptualization:** Junaid Nasir Qureshi.

**Methodology:** Junaid Nasir Qureshi, Muhammad Shoaib Farooq.

**Supervision:** Muhammad Shoaib Farooq.

**Writing – original draft:** Junaid Nasir Qureshi.

**Writing – review & editing:** Junaid Nasir Qureshi, Muhammad Shoaib Farooq.

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
