## [Decision Letter · Decision Letter 0]

6 Dec 2023

PONE-D-23-38879ChainAgile: A Framework for the Improvement Of Scrum Agile Distributed Software Development Based On BlockchainPLOS ONE

Dear Dr. Farooq,

Thank you for submitting your manuscript to PLOS ONE. After careful consideration, we feel that it has merit but does not fully meet PLOS ONE’s publication criteria as it currently stands. Therefore, we invite you to submit a revised version of the manuscript that addresses the points raised during the review process.

We look forward to receiving your revised manuscript.

Kind regards,

Mohammed Shuaib

Academic Editor

PLOS ONE

Journal Requirements:

Reviewers' comments:

Reviewer's Responses to Questions

**Comments to the Author**

1. Is the manuscript technically sound, and do the data support the conclusions?

Reviewer #1: Yes

Reviewer #2: Yes

2. Has the statistical analysis been performed appropriately and rigorously? 

Reviewer #1: Yes

Reviewer #2: No

3. Have the authors made all data underlying the findings in their manuscript fully available?

Reviewer #1: Yes

Reviewer #2: Yes

4. Is the manuscript presented in an intelligible fashion and written in standard English?

Reviewer #1: Yes

Reviewer #2: No

5. Review Comments to the Author

Reviewer #1: 1) Authors should add motivation and contribution in the introduction section

2) Author should make one table in related study with existing work and their limitations?

3) How the proposed model is beneficial in decentralized system?

4) All the table must be improved and the text within the table must be aligned properly.

5) The grammar and typos error must be taken care

6) Author should add advantages and disadvantages of the proposed model.

7) Is the proposed system secure enough and sutainable to apply in distributed environment. If yes, justify it, how the proposed model is different from mentioned research below.

a. Blockchain-Based Authentication and Explainable AI for Securing Consumer IoT Applications." IEEE Transactions on Consumer Electronics

b. Blockchain and Deep Learning for Secure Communication in Digital Twin Empowered Industrial IoT Network. IEEE Transactions on Network Science and Engineering.

c. Blockchain and deep learning empowered secure data sharing framework for softwarized uavs. In 2022 IEEE International Conference on Communications Workshops (ICC Workshops) (pp. 770-775). IEEE.

d. Fog intelligence for secure smart villages: Architecture, and future challenges. IEEE Consumer Electronics Magazine.

Reviewer #2: Introduction:

there is a need to explain the blockchain's role and give an understanding of its importance.

The contribution is written well to understand the proper achievements of the authors.

Figure 1 very simply presents the waterfall model with iteration after the completion of a module as a sprint. It’s not suitable to explain the agile lifecycle. Reconsider to improve and redesign properly according to process.

Figure 2 is also not good enough to explain the proposed hybrid model of agile with the combination of blockchain. What the author wants to show in figure 2 exactly.

There should be a paragraph at the bottom of the introduction that organizes the rest of the sections of the article.

Related work:

Literature should be section-wise when discussing hybrid techniques with multiple technologies.

Materials and methods:

It looks like a conceptual framework instead of concrete properly implemented. This section explains figures but does not give a proper deep understanding.

What is the exact role of IPFS? IPFS is used for storing metadata as a document to decentralize, but the author didn’t explain the exact role where they used IPFS and how they used it.

The author didn’t explain the research methodology of the study. It is essential to understand how the research was conducted, including the approach, data collection methods, and any relevant procedures.

This section needs to be revised properly; it’s not showing impressive work.

Experimental and results:

The author didn’t explain the environment of the blockchain and how they implemented their experiments. The author only said they used Python language and Replit IDE, Replit is used for different types of multiple languages, so why do they need to use it, what’s the link with blockchain?

Postman is just for testing the API call, what do they want to achieve by Postman?

How did they evaluate the cost/gas of transactions when executing the smart contract with Ethereum technology? Nothing is there to see.

There is a lack of explanation regarding the role and implementation of blockchain technology within the study. It is important to provide a clear explanation of the role of blockchain technology in the context of the study's implementation.

6. PLOS authors have the option to publish the peer review history of their article (what does this mean?). If published, this will include your full peer review and any attached files.

Reviewer #1: No

Reviewer #2: **Yes: **Abdul Razzaq

---

## [Author Response · Author response to Decision Letter 0]

22 Jan 2024

Submission ID: PONE-D-23-38879

Original Article Title: “ChainAgile: A Framework for the Improvement Of Scrum Agile Distributed Software Development Based On Blockchain”

To: PLOS ONE

Dear Editor,

First, we are extremely grateful to the editor and all the reviewers who conducted a rigorous and constructive review of our article number PONE-D-23-38879. Certainly, all the review points were very well directed, and we managed to improve our manuscript a great deal and had to work a lot to address all the changes suggested by the reviewers.

We have incorporated all the suggested reviews and comments provided by the reviewers and made a careful revision of the manuscript. We have tried our level best to improve this manuscript by organizing and adding more references to the available relevant literature, and by presenting the material more constructively and effectively. This has certainly raised the quality of this research work and will make it a valuable contribution to the community. We hope that we have managed to make the required changes suggested by the valuable reviewers.

Please see our responses in line with the editor’s and reviewers’ concerns in green color.

Best regards,

Junaid et al.

Reviewer#1, Concern # 1: Authors should add motivation and contribution in the introduction. 

Author response: Thank you for your valuable comment. As per the reviewer's suggestion, we have highlighted the motivation and contribution in the manuscript. 

Author action: We have updated and revised the introduction section of the manuscript. 

Reviewer#1, Concern # 2: Author should make one table in related study with existing work and their limitations?

Author response: Thank you for your valuable comment. According to the reviewer's suggestion, we have already designed the table of comparison in the manuscript which compares the existing systems with our proposed model against the filters which are blockchain, communication, coordination, security, scalability, prioritization, review/backlog and testing. The new research articles also compared in this table. We have added the limitations’ part against each reference compared in related work of the manuscript in related work section.

Author action: We have revised and updated Table 1 in the manuscript and added the limitations against each reference in the manuscript when discussing the related work.________________________________________

Reviewer#1, Concern # 3: How the proposed model is beneficial in decentralized system?

Author response: Thank you for your valuable comment. As per the reviewer's suggestion, we have highlighted the proposed model benefits in a decentralized systems in the updated manuscript. 

Author action: We have revised and updated the introduction section of manuscript. 

Reviewer#1, Concern # 4: All the tables must be improved and the text within the table must be aligned properly.

Author response: Thank You for the valuable comment. We have updated all the tables and all the content in the manuscript with alignment as well. 

Author action: We updated the manuscript by reviewing and adjusting the text and alignment throughout the paper to ensure improvement and clarity.

Reviewer#1, Concern # 5: All The grammar and typos error must be taken care 

Author response: We appreciate the reviewer's keen observation regarding grammar language typos error. As per the suggestion it is thoroughly studied and corrected the grammar, language and typos error in the manuscript. 

Author action: We updated the manuscript by reviewing and adjusting the grammar, language and typos error throughout the manuscript to ensure grammatical accuracy and clarity.

Reviewer#1, Concern # 6: Author should add advantages and disadvantages of the proposed model.

Author response: We appreciate the reviewer's keen observation regarding proposed model advantages and disadvantages. As per the suggestion, we have added at the end of the introduction section the proposed model role and its advantages.

Author action: We updated the manuscript by adding the proposed model role and its advantages in the introduction section of the document.

Reviewer#1, Concern # 7: Is the proposed system secure enough and sustainable to apply in distributed environment? If yes, justify it, how the proposed model is different from mentioned research below.

a. Blockchain-Based Authentication and Explainable AI for Securing Consumer IoT Applications." IEEE Transactions on Consumer Electronics

b. Blockchain and Deep Learning for Secure Communication in Digital Twin Empowered Industrial IoT Network. IEEE Transactions on Network Science and Engineering.

c. Blockchain and deep learning empowered secure data sharing framework for softwarized uavs. In 2022 IEEE International Conference on Communications Workshops (ICC Workshops) (pp. 770-775). IEEE.

d. Fog intelligence for secure smart villages: Architecture, and future challenges. IEEE Consumer Electronics Magazine.

Author response: We appreciate and are grateful for the reviewer's keen observation regarding literature review and specifically mentioning some specified articles with us. According to the concern yes are framework is suitable for secure and sustainable distributed environment and how are proposed model is different is expressed in related work section by comparing all mentioned articles. 

Author action: We updated and revised the manuscript by reviewing and comparing the mentioned articles in our manuscript related work section.

Reviewer #2: Introduction

Reviewer#2, Concern # 1: There is a need to explain the blockchain's role and give an understanding of its importance.

Author response: Thank you for your valuable comment. As per the reviewer's suggestion, we have highlighted the proposed model role with blockchain and its benefits in a decentralized systems in the manuscript. 

Author action: We have updated and revised the manuscript by modifying the proposed framework description and Materials and Methods section of the manuscript.

Reviewer#2, Concern # 2: The contribution is written well to understand the proper achievements of the authors.

Author response: Thank you for your valuable and must appreciated comments. The reviewer's valuable input provides clarity of understanding for the author.

Author action: We thank you, for your admirable and valuable comment. 

Reviewer#2, Concern # 3: Figure 1 very simply presents the waterfall model with iteration after the completion of a module as a sprint. It’s not suitable to explain the agile lifecycle. Reconsider to improve and redesign properly according to process.

Author response: We appreciate the reviewer’s insightful suggestion. To provide a more specific detailed diagram we have incorporated and changed the diagram which shows the agile scrum’s sprint lifecycle.

Author action: To address the reviewer's comment, we have updated the manuscript by providing a new diagram which reflects the agile scrum’s sprint lifecycle.

Reviewer#2, Concern # 4: Figure 2 is also not good enough to explain the proposed hybrid model of agile with the combination of blockchain. What the author wants to show in figure 2 exactly.

Author response: Thank you for your valuable suggestion. We have improved the diagram, and its explanation is added in the updated manuscript, which reflects the basic idea of the diagram. We provide how the blockchain features help our proposed model framework.

Author action: We updated the manuscript by adding an explanation of Blockchain features of Figure 2 and redesigned Figure 2 as well. ________________________________________

Reviewer#2, Concern # 5: There should be a paragraph at the bottom of the introduction that organizes the rest of the sections of the article.

Author response: Thank you for the insightful comment. We have updated the manuscript and added the paragraph at the end of the introduction which tells the rest of the sections in this article.

Author action: We have modified and updated the manuscript by adding the paragraph at the end of the introduction section, which highlights the rest of the article sections.

Related work

Reviewer#2, Concern # 6: Literature should be section-wise when discussing hybrid techniques with multiple technologies.

Author response: We appreciate the reviewer's feedback concerning the literature review section wise. We have updated the manuscript and its related work portion in different section wise. This will explain, distinguish provides clarity and also articulate the novelty about our proposed framework. 

Author action: We updated the manuscript by revising the complete related work literature section. ________________________________________

Materials and methods

Reviewer#2, Concern # 7: It looks like a conceptual framework instead of concrete properly implemented. This section explains figures but does not give a proper deep understanding. 

Author response: Thank you for your insightful comment. In response to your feedback, we have enhanced and modified the materials and methods section in depth and detail. The proposed framework is explained, and its comprehensive description is shown in Figure 7. Proposed framework is explain in six layers of scrum agile agreement, requirement elicitation, prioritization, design and development, testing and payment. The blockchain architecture layer wise concrete mechanism how to implement well described and depicts in Figure 8. Layered architecture with blockchain integration in scrum agile distributed software development explained layer wise and how it works shown in Figure 9. How all layered wise architectural processes of blockchain integration with scrum agile works and communicate with each other in distrusted agile software development shown in Figure 10. All JSON objects work on different layers with different functionality are described from JSON objects 1 to JSON object 14. The ChainAgile ecosystem of blockchain integration explained in stages and showed in Figure 12, how blockchain combines with scrum agile distributed environment. Blockchain transaction flow explained in the flow chart diagram with its description in Figure 16, how a transaction occurs in ChainAgile. Therefore, we have tried to describe and explain the framework concrete implementation how it works with the combination of blockchain integrated with scrum agile which provides the more insight into the flow of the study.

Author action: We have updated and modified the manuscript by adding different subsection details under the materials and methods section.

Reviewer#2, Concern # 8: What is the exact role of IPFS? IPFS is used for storing metadata as a document to decentralize, but the author didn’t explain the exact role where they used IPFS and how they used it.

Author response: Thank you for your valuable comment. To address the reviewer's comment, we have updated the Materials and Methods, proposed framework, introduction, and discussion sections respectively where the role of IPFS is explained and described. IPFS is used to solve the scalability problem of blockchain for storing the records of customers, developers, and their communication. IPFS to address scalability issues. We have provided a more detailed description of how IPFS is integrated, specifically highlighting the utilization of the IPFS Desktop App.

Author action: We have updated and revised the manuscript by modifying the proposed framework description, Materials and Methods section, introduction, and discussion sections respectively.

Reviewer#2, Concern # 9: The author didn’t explain the research methodology of the study. It is essential to understand how the research was conducted, including the approach, data collection methods, and any relevant procedures.

Author response: Thank you for your valuable comment, as per the reviewer's suggestion we agree with the reviewer, added the description and diagrams for research methodology. We added Figure 3 Research methodology, Figure 4 Digital Library Search and Figure 5 Year wise articles count for this and make the table 2 search string for the detail needed for research methodology. 

Author action: We have updated and revised the manuscript by adding the research methodology section under the materials and methods section.

Experimental and results

Reviewer#2, Concern # 10: The author didn’t explain the environment of the blockchain and how they implemented their experiments. The author only said they used Python language and Replit IDE, Replit is used for different types of multiple languages, so why do they need to use it, what’s the link with blockchain?

Author response: Thank you for your valuable comment. As per the reviewer's suggestion we have explained blockchain understandings and integration under the Materials and Method Section. Experiment implementation and its process explained under the experiment and result Cost/Gas section with some changes in experiment. The performance and evaluation against the experiment under the ChainAgile framework section explains the details respective to Ethereum Cost/Gas, longest chain retrieval and chains block size. The Replit IDE is used to generate the chain and for the creation of blocks in Python. The Latency and Block size results retrieved as software as service. The software as service is cloud based and used to perform the experiment with blockchain integrated environment.

Author action: We have updated the revised the manuscript by updating and describing the experiment with changes, its result in form of performance evaluation under Materials and Method , experiment and result Cost/Gas, and performance and evaluation ChainAgile framework sections respectively.

Reviewer#2, Concern # 11: Postman is just for testing the API call, what do they want to achieve byPostman?

Author response: Thank you for your valuable comment. The Postman is used for testing purpose of API Calls. We used this to get and post the http request to the network. The Postman is employed for all HTTP requests used during the experiment engaged with the blockchain. Configured the transactions in HTTP request forms using the Postman tool, to validate the effectiveness of ChainAgile proposed framework.

Author action: We updated the manuscript by describing the postman process under Materials and methods section.

Reviewer#2, Concern # 12: How did they evaluate the cost/gas of transactions when executing the smart contract with Ethereum technology? Nothing is there to see.

Author response: We appreciate your insightful suggestion. In response to your recommendation regarding the cost/gas of transactions we designed with some changes the experiments and its results with description mentioned in Experiment and results section. Figure 22 is added to show the block addition in the network and Figure 23 shows the cost/gas of transactions. acknowledgment of implementation challenges and limitations, we have incorporated a dedicated section addressing these parts within the "Propose Farmwork" section. The added content explores the challenges associated with scalability, limitations of IPFS, integration complexity, complexity of smart contracts, and finally user adoption and training.

Author action: We updated the manuscript by adding the ‘cost/gas of transactions’ under the section of experiments and results.________________________________________

Reviewer#2, Concern # 13: There is a lack of explanation regarding the role and implementation of blockchain technology within the study. It is important to provide a clear explanation of the role of blockchain technology in the context of the study's implementation.

Author response: Thank you for the suggestion. We have enhanced and incorporated the detail discussion on the role and implementation of blockchain technology within the study by providing the blockchain integration understandings, its benefits and advantages, how blockchain is integrated with scrum agile distributed environment, how the proposed model works with blockchain, its smart contracts reflect the JSON objects, blockchain role and its contribution in the proposed ChainAgile framework. 

Author action: We updated the manuscript by providing the details under Introduction, Materials and methods and experiment and results section respectively.

---

## [Decision Letter · Decision Letter 1]

9 Feb 2024

ChainAgile: A Framework for the Improvement Of Scrum Agile Distributed Software Development Based On Blockchain

PONE-D-23-38879R1

Dear Dr. Farooq,

We’re pleased to inform you that your manuscript has been judged scientifically suitable for publication and will be formally accepted for publication once it meets all outstanding technical requirements.

Kind regards,

Mohammed Shuaib

Academic Editor

PLOS ONE

Additional Editor Comments (optional):

Reviewers' comments:

Reviewer's Responses to Questions

**Comments to the Author**

1. If the authors have adequately addressed your comments raised in a previous round of review and you feel that this manuscript is now acceptable for publication, you may indicate that here to bypass the “Comments to the Author” section, enter your conflict of interest statement in the “Confidential to Editor” section, and submit your "Accept" recommendation.

Reviewer #1: All comments have been addressed

2. Is the manuscript technically sound, and do the data support the conclusions?

Reviewer #1: Yes

3. Has the statistical analysis been performed appropriately and rigorously? 

Reviewer #1: Yes

4. Have the authors made all data underlying the findings in their manuscript fully available?

Reviewer #1: Yes

5. Is the manuscript presented in an intelligible fashion and written in standard English?

Reviewer #1: Yes

6. Review Comments to the Author

Reviewer #1: All comments are incorporated successfully. Needs more work on related study and framing of conclusion.

7. PLOS authors have the option to publish the peer review history of their article (what does this mean?). If published, this will include your full peer review and any attached files.

Reviewer #1: **Yes: **RANDHIR KUMAR

---

## [Editor Report · Acceptance letter]

23 Feb 2024

PONE-D-23-38879R1 

PLOS ONE

Dear Dr. Farooq, 

I'm pleased to inform you that your manuscript has been deemed suitable for publication in PLOS ONE. Congratulations! Your manuscript is now being handed over to our production team.

Kind regards, 

on behalf of

Dr. Mohammed Shuaib 

Academic Editor

PLOS ONE